# Structural characterization of antibody-responses following Zolgensma treatment for AAV capsid engineering to expand patient cohorts

Mario Mietzsch [1] ✉, Jane Hsi [1], Austin R. Nelson[1], Neeta Khandekar[2], Ann-Maree Huang[2], Nicholas JC Smith [3,4], Jon Zachary[1], Lindsay Potts [1], Michelle A. Farrar [5,6], Paul Chipman [7], Mohammad Ghanem[8], Ian E. Alexander [2,9], Grant J. Logan[2], Juha T. Huiskonen [8] & Robert McKenna [1] ✉

Monoclonal antibodies are useful tools to dissect the neutralizing antibody response against the adeno-associated virus (AAV) capsids that are used as gene therapy delivery vectors. The presence of pre-existing neutralizing antibodies in large portions of the human population poses a significant challenge for AAV-mediated gene therapy, primarily targeting the capsid leading to vector inactivation and loss of treatment efficacy. This study structurally characterizes the interactions of 21 human-derived neutralizing antibodies from three patients treated with the AAV9 vector, Zolgensma®, utilizing high-resolution cryo-electron microscopy. The antibodies bound to the 2-fold depression or the 3-fold protrusions do not conform to the icosahedral symmetry of the capsid, thus requiring localized reconstructions. These complex structures provide unprecedented details of the mAbs binding interfaces, with many antibodies inducing structural perturbations of the capsid upon binding. Key surface capsid amino acid residues were identified facilitating the design of capsid variants with antibody escape phenotypes. These AAV9 capsid variants have the potential to expand the patient cohort to include those that were previously excluded due to their pre-existing neutralizing antibodies against the wtAAV9 capsid, and the possibly of further treatment to those requiring redosing.

Adeno-associated viruses (AAVs) have become the leading therapeutic gene delivery vector, as they are non-pathogenic, possess the ability to package and express therapeutic genes in a wide range of cell and tissue types[1–3]. They can mediate long-term correction of monogenic diseases and have been shown to be effective in numerous clinical trials. These successes have resulted in the approval of currently eight AAV-based biologics, including Zolgensma, an AAV9-based therapy for the treatment of spinal muscular atrophy[4–7]. The AAV vectors are composed of non-enveloped $T = 1$ icosahedral capsids, consisting of 60 viral proteins (VP), and package single-stranded DNA transgene cassettes[8].

The structures of several AAV capsids have been determined by X-ray crystallography and/or cryo-electron microscopy (cryo-EM)[8–10]. All AAV capsids VPs consist of a conserved jelly-roll core and variable

surface loops/regions (VRs)[8]. The capsids are assembled via 2-, 3-, and 5-fold symmetry-related VP interactions[11]. The icosahedral 5-fold axes are surrounded by five loops that form the cylindrical channels which connect the interior to the exterior environment. Another characteristic feature of the capsids is protrusions surrounding the 3-fold axes. At the 2-fold axes and surrounding the 5-fold channels, the AAV capsids possess depressions that are separated by raised regions termed 2/5-fold walls. Structural variations between different AAV capsids arise from differences in amino acid (aa) sequences, resulting in alternative receptor usage leading to distinct host and/or tissue tropisms and determining their antigenic profiles[12,13].

A major challenge for the utilization of AAV vectors in the clinic is the presence of pre-existing neutralizing antibodies (NAbs) in a large percentage of the human population[14–16]. These antibodies originate from prior exposure to naturally circulating AAVs, primarily targeting the capsid, leading to vector inactivation and loss of treatment efficacy. As a result, patients with these pre-existing antibodies and exceeding defined antibody titer thresholds are excluded from receiving these therapeutics[17,18]. One strategy to circumvent these antibodies is the engineering of capsid variants that evade these pre-existing NAbs. This approach requires the identification of the antibody epitopes on the capsid surfaces. For this purpose, the fragment antigen binding (Fab) domains of the immunoglobulins (Ig) are incubated to the capsid and the complex structure is determined by cryo-EM. A Fab is composed of the light chain and the N-terminal half of the heavy chain. Each chain possesses three complementarity-determining regions (CDRs), which are responsible for recognizing specific antigens[19].

Initially, the resolutions of AAV capsid-antibody cryo-EM maps were low (~5–20 Å)[8,20,21]. However, recent advancements of the cryo-EM technology have enabled the analysis of the interaction between the antibody and the capsid surface at atomic resolution[22]. The identified interacting residues form the basis for AAV capsid engineering efforts by rational design or by directed evolution[23]. Modifications of capsid aa have been shown to result in capsid variants capable of escaping the characterized antibodies in vitro and in vivo[20–22,24]. To date, this structural mapping approach has relied primarily on mouse monoclonal antibodies (mAbs) to simulate the human antibody response against the AAV capsids. A recent study obtained 21 individual human mAbs from three patients following Zolgensma® treatment, which utilizes the capsids of AAV serotype 9[25]. The identification of their binding sites to the AAV9 capsid at low resolution showed that the majority of human mAbs bound to the 2-fold region of the capsid, whereas mouse mAbs preferentially bound to the 3-fold protrusions[20,25].

This study identifies the interacting residues of neutralizing human mAbs using high-resolution cryo-EM. However, due to mAbs bound to the icosahedral symmetry axes, 3D-reconstructions of the complexes with imposed icosahedral symmetry result in "blurred" densities for the Fabs, preventing effective model building. To overcome the symmetry mismatches, localized reconstructions with symmetry relaxation are utilized, enabling the deconvolution of the alternative binding modes of the Fabs. Utilizing this structural information, key residues in the AAV9 capsid are modified to generate capsid variants escaping up to 18 of the 21 mAbs at high antibody titers. These AAV capsid variants have the potential to expand the patient cohort treatable with AAV vectors to include patients that were previously excluded due to their pre-existing antibodies and open the possibility of re-administration of the vector if such a need arises.

## Results
### Icosahedral averaging only resolves Fabs binding away from the symmetry axes
For a detailed understanding of the interaction between the human mAbs and the AAV9 capsids, high-resolution cryo-EM data were collected for each of the capsid-antibody complexes. Utilizing standard 3D-reconstruction protocols by imposing icosahedral symmetry using ~43,000–366,000 particles resulted in cryo-EM maps at resolutions ranging from 1.88 to 3.27 Å (Supplementary Table 1). For each map, the density contributed by the bound Fab was detected at the previously determined capsid regions[25]. At the 2/5-fold wall of the capsid, Fab1-2 was resolved at a resolution of 2.61 Å (Fig. 1a). In the map, the capsid, the variable heavy ($V_H$) and light ($V_L$) chains were well ordered and allowed the building of reliable models. In contrast, the constant regions of the Fabs were less ordered and not observed at a sigma ($\sigma$) threshold level of 2.0, likely due to a higher level of flexibility as they do not participate in the complex formation. Around the 5-fold axis, Fab1-6 and Fab2-7 were resolved at a resolution of 2.33 and 2.18 Å (Supplementary Table 1) but adopting differential binding modes. While Fab1-6 bound closer to the 5-fold channel, Fab2-7 bound mostly to the depressed region surrounding the 5-fold channel, leaning towards the 2/5-fold wall (Fig. 1b). In both maps the $V_H$ and $V_L$ chains were well ordered, enabling the building of atomic models, except for some regions of the $V_H$ chain of Fab1-6 located close to the 5-fold channel. The latter region was less structurally ordered, and the built model of the Fab resulted in clashes with the neighboring, symmetry-related Fabs. However, the disordered density observed for Fab1-6 was only minor in comparison to all 2-fold and 3-fold binding Fabs. The complexes with the two 3-fold binding Fabs, Fab1-1 and Fab3-4, were reconstructed to 2.62 and 3.27 Å resolution. In both cryo-EM maps, the capsid was well ordered. However, the density from the Fabs observed directly above the 3-fold symmetry axis was disorganized and could not be used to build atomic models (Fig. 1c). This lack of structural ordering is the result of only a single Fab occupying the space at the 3-fold axis in one of three binding modes. Consequently, the bound Fab does not conform to the symmetry of the capsid, and icosahedral averaging imposed during the 3D reconstruction blurred the density of the Fab. A similar situation was observed with the 2-fold binding Fabs. The complexes of the 16 Fabs binding to this region of the capsid were reconstructed to a resolution of 1.88–2.84 Å resolution (Supplementary Table 1). In each map, the capsid was well ordered, while the densities for the Fabs above the 2-fold axis could not be interpreted for model building (Fig. 1d). As for the 3-fold antibodies, only a single Fab bound to the 2-fold region, in two potential binding modes, resulting in distorted densities during icosahedral averaging.

### Localized reconstruction resolves Fabs bound close to the symmetry axes
The visualization of asymmetric features in otherwise icosahedral structures, such as the Fabs binding near or at the capsids' icosahedral symmetry axes, using traditional cryo-EM reconstruction procedures has been challenging, as their information is lost due to averaging of the particle. However, standard asymmetric reconstruction is unfeasible here as the Fab binding modes are independent at each axis, resulting in a vast number of structural classes (~$3^{20}$ possible permutations for the 3-fold binder)[26]. Thus, the localized reconstruction method was used in combination with symmetry relaxation to structurally characterize the 2-, 3-, or 5-fold regions, independent of the entire capsid, to reconstruct the Fabs not conforming to the icosahedral symmetry of the capsid[26–28]. In this process, sub-particle maps were generated comprising only the region around the symmetry axis of choice. For the sixteen 2-fold binding Fabs, this strategy resulted in maps reconstructed to 2.54–3.56 Å resolution (Supplementary Table 1). The localized reconstructions with C2 symmetry relaxation confirmed that only a single Fab is bound at the 2-fold region (Fig. 2a). In all maps, the $V_H$ and $V_L$ chains are well-ordered, enabling the building of atomic models. An exception was Fab2-6 with a well-ordered $V_H$ chain but a poorly ordered $V_L$ chain, indicating that the $V_L$ may not participate in capsid binding. Some maps also resolved the constant heavy ($C_H$) and light chain ($C_L$) of the Fabs such as Fab2-1

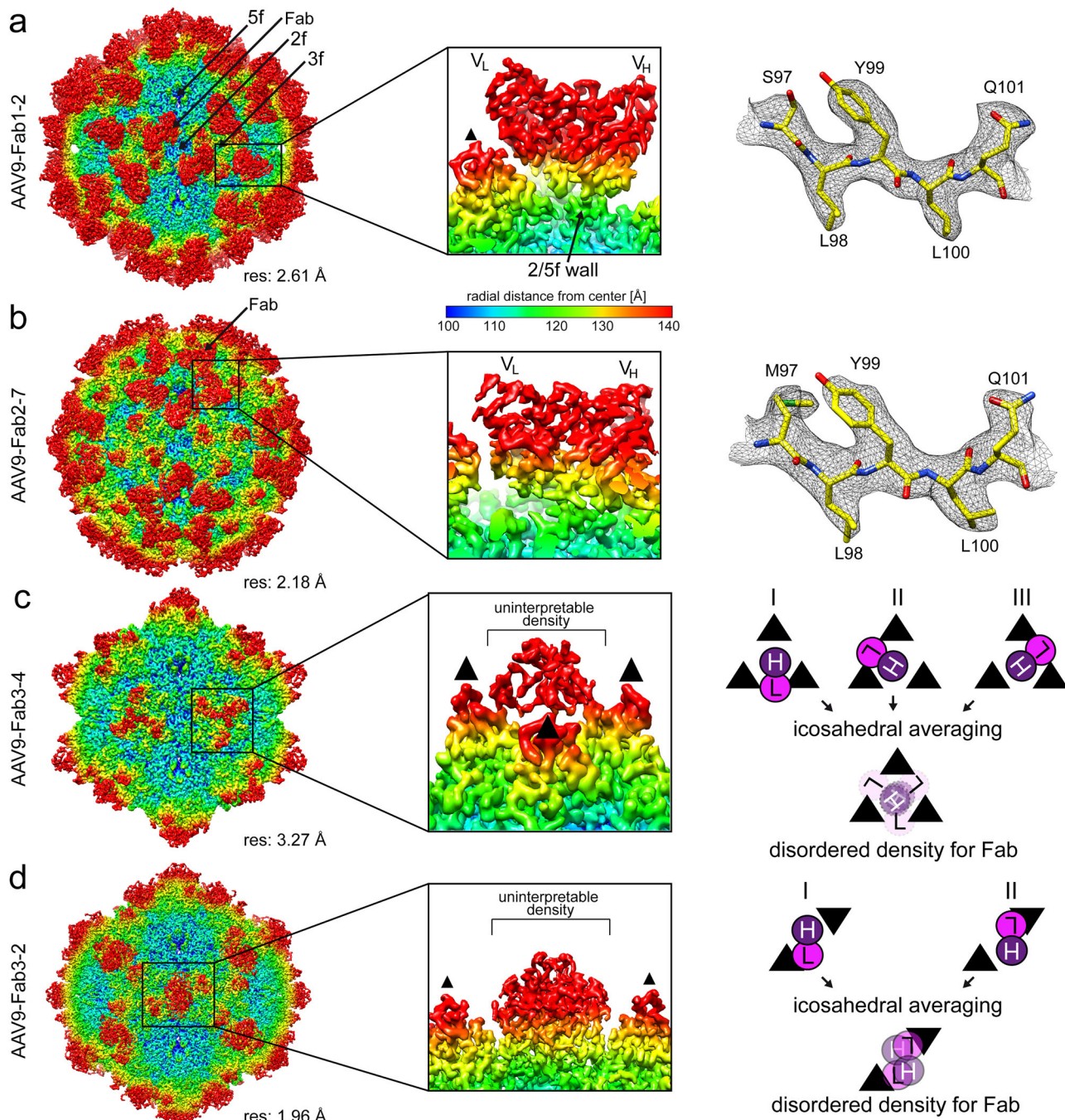

**Fig. 1 | Icosahedral reconstruction of the AAV9-Fab complexes. a** Surface density map of the cryo-reconstructed AAV9-Fab1-2 complex. The view is down the icosahedral 2-fold axis, the map is contoured at a sigma ($\sigma$) threshold level of 2.0. The map is colored according to radial distance from the particle center (blue to red), as indicated by the scale bar. The icosahedral 2-, 3-, and 5-fold axes are indicated. The estimated resolution, determined at an FSC threshold value of 0.143, is shown below the map. The binding of the Fab on top of the 2/5-fold wall is shown as a close-up image in the center panel, with the variable heavy ($V_H$) and variable light ($V_L$) chains labeled. To the right, a representative stretch of amino acid residues modeled for the heavy chain of the Fab are shown inside the cryo-EM density map. The amino acid residues are labeled and shown as stick representations and colored according to atom type: C = yellow, O = red, N = blue, S = green. **b** Depiction of the AAV9-Fab2-7 complex as in (**a**) with the Fab binding around the 5-fold symmetry axis. **c** Surface density map of the AAV9-Fab3-4 complex as in (**a**). The Fab binds to the center of the 3-fold symmetry axis. Due to the imposed symmetry during the reconstruction process the density of the Fab is blurred and cannot be used to generate an atomic model. The illustration on the right gives a rationale for the blurring of a non-icosahedral Fab (H: heavy chain [purple], L: light chain [pink]) bound to an icosahedral capsid during icosahedral averaging. **d** Depiction of the AAV9-Fab3-2 complex as in (**c**) with the Fab binding to the center of the 2-fold symmetry axis. The 3-fold protrusions are indicated by small black triangles. All density map images were generated using UCSF-Chimera[61].

(Fig. 2a). For the two 3-fold binding Fabs, Fab1-1 and Fab3-4, localized reconstruction also resolved the single Fab at the 3-fold axis at a resolution of 3.31 and 3.73 Å (Supplementary Table 1 and Fig. 2b). In the case of Fab1-1, the $V_H$ and $V_L$ chains are well ordered, whereas for Fab3-4 only the $V_H$ chain is well-ordered, indicating that capsid binding is mainly mediated by $V_H$. Lastly, for the 5-fold Fabs, the AAV9-Fab1-6 complex was processed with localized reconstruction, resulting in a 3.47 Å resolution map. In contrast to single Fabs binding at the 2- and 3-fold, the cryo-EM map revealed two bound Fabs with well-ordered $V_H$ and $V_L$ chains, surrounding the 5-fold axis in an opposing manner,

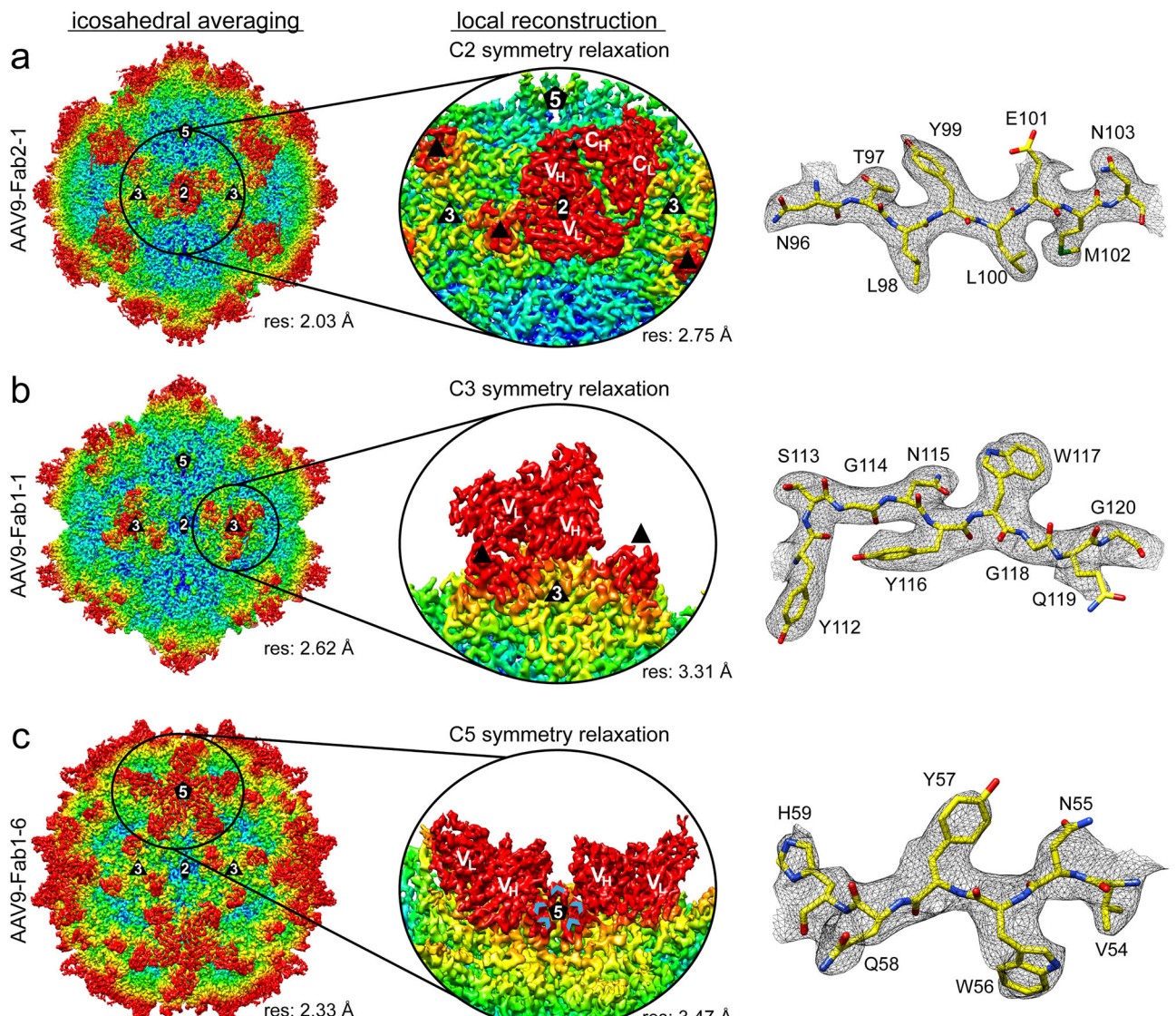

**Fig. 2 | Localized reconstruction of the AAV9-Fab complexes. a** Surface density map of the icosahedral-reconstructed AAV9-Fab2-1 complex. The map is colored according to radial distance from the particle center (blue to red), as indicated by the scale bar in Fig. 1. To the right, a density map of the 2-fold region is shown, derived from localized reconstruction with C2 symmetry relaxation. The variable/constant heavy ($V_H/C_H$) and variable/constant light ($V_L/C_L$) chains are labeled. A representative stretch of amino acid residues modeled for the heavy chain of the Fab is shown inside the localized reconstructed map. The amino acid residues are labeled and shown as stick representations and colored according to atom type:

C = yellow, O = red, N = blue, S = green. **b** Depiction of the AAV9-Fab1-1 and (**c**) Fab1-6 complex as in (**a**). For the localized reconstruction of the 3-fold region C3 symmetry relaxation and the 5-fold region C5 symmetry relaxation were applied, respectively. The maps are colored according to radial distance from the particle center (blue to red), as in Fig. 1. The icosahedral 2-, 3-, and 5-fold axes are indicated as ovals, triangles, and pentagons, respectively. The estimated resolution is shown below the map. The 3-fold protrusions and the positions of the VR-II loops around the 5-fold axis are indicated as black triangles and blue arrowheads, respectively.

thereby avoiding clashes with Fabs bound at the nearest 5-fold related capsid monomer (Fig. 2c).

**The 2-fold binding Fabs exhibit four distinct binding modes**

The atomic models for the Fabs allowed the comparison of their binding patterns to the AAV9 capsid. The sixteen Fabs binding to the 2-fold axis could be structurally sub-classified into 4 groups (Fig. 3a). In group A, containing Fab2-1, 2-3, 2-4, 2-6, 3-1, 3-2, and 3-5, the $V_H$ chain of the Fab binds perpendicular to the 2-fold axis with its CDRs entering the depression. Compared to $V_H$, the $V_L$ chain is shifted toward the 2/5-fold wall and rotated ~90° with its CDRs binding to the side of the 3-fold protrusions. The group B Fabs, Fab1-5, 2-2, 3-6, and 3-7, are rotated ~20° relative to group A, shifting the $V_L$ chain away from the side of the 3-fold protrusions. In group C, comprising of Fab1-3, 1-4,

and 2-5, it is the $V_L$ chain that is situated above the 2-fold axis. The $V_H$ chain is shifted towards the 2/5-fold wall binding to the side of the 3-fold protrusions with CDR1/2, whereas the long CDR3 loop enters the 2-fold depression. This binding mode resembles the group A Fabs with the position of the heavy and light chains swapped. However, in both groups the CDR3 loop of the $V_H$ chain occupies approximately the same space in the 2-fold depression. In group D, Fab1-7 and 3-3, the $V_H$ chain is located on top of the 2-fold axis with CDR1 and CDR2 entering the 2-fold depression while CDR3 binds to the side of the 3-fold protrusions. Unlike for group A, the $V_L$ chains are elevated higher from the capsid, with their CDRs making contact with the top of the 3-fold protrusions. To investigate whether the grouping can be correlated to the Fabs aa sequence, phylogenetic trees for the $V_H$ and $V_L$ chains were generated. In the analysis, the $V_H$ chains did not form clusters for the

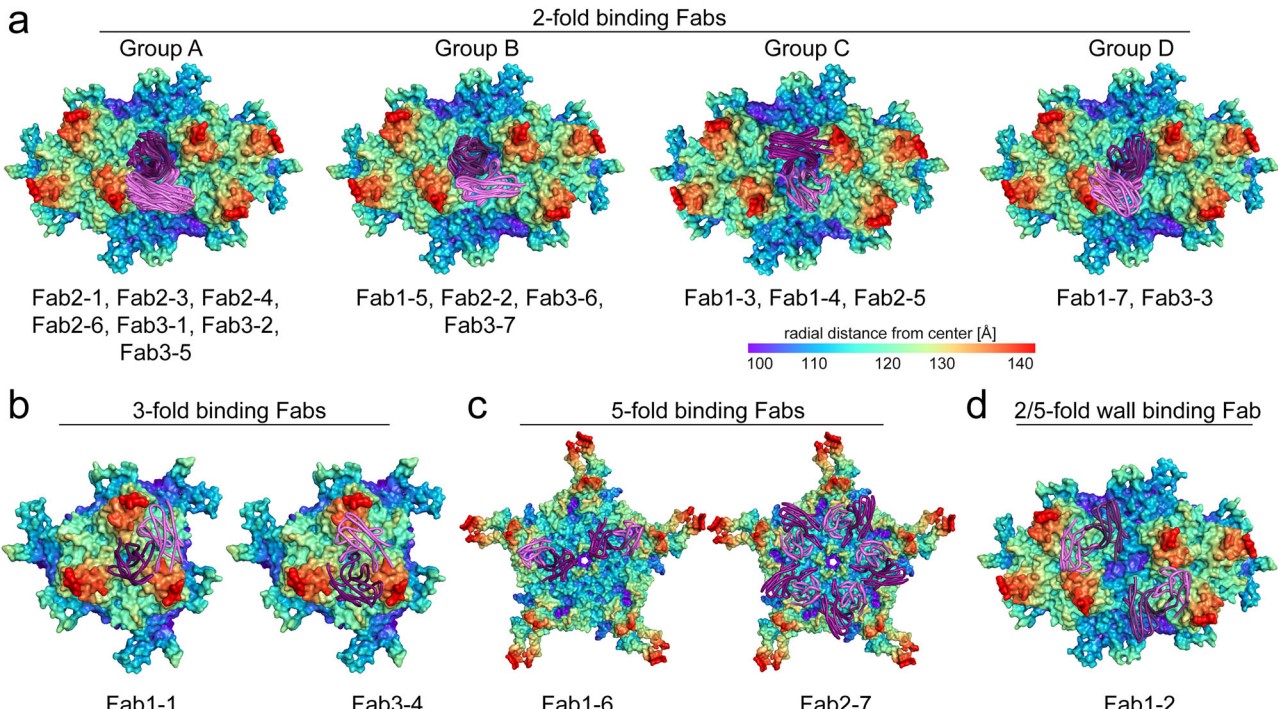

**Fig. 3 | Binding mode of the Fabs to the AAV9 capsid. Radial-colored surface representations of the AAV9 capsid are shown. a** For double-trimers with the 2-fold symmetry axis in the center, **b** for trimers with the 3-fold symmetry axis in the center, **c** for pentamers with the 5-fold symmetry axis in the center, and **d** for double-trimers as in (**a**). The surface representations are colored according to radial distance from the capsid's center (blue to red), as indicated by the scale bar. The structures of the $V_H$ (purple) and $V_L$ (pink) chains of the Fabs are shown as ribbon diagrams. The 2-fold binding Fabs are classified into 4 groups (A, B, C, and D) based on their interaction with the AAV9 capsid. The Fabs belonging to the groups are listed below each surface representation.

individual Fab groups except for group D (Supplementary Fig. 1a). A common feature of the group D Fabs is their much shorter CDR3$_H$ loop (Supplementary Fig. 1b), which does not penetrate the 2-fold depression. In contrast, the group A Fabs possess, on average, the largest CDR3 loop for the $V_H$ chain. This is probably the result of the chain slightly shifting towards the 2/5-fold wall and the need to still enter the 2-fold depression. The remaining five CDRs of the Fab do not show significant length differences as previously reported[25]. The phylogenetic analysis of the $V_L$ chains showed a clustering of group B and most of group A Fabs (Supplementary Fig. 1c), suggesting a potential role of the light chain for the binding mode of the Fab.

The Fabs, 1-1 and 3-4, are located differently in the 3-fold region. While the $V_H$ chain of Fab1-1 is positioned directly above the 3-fold symmetry axis, $V_H$ of Fab3-4 is located further away from the 3-fold axis between two protrusions (Fig. 3b). Despite their different locations, the CDR3 loop of both Fabs enters the depression at the center of the 3-fold axis. To achieve this, the CDR3 of Fab3-4 is three aa longer. The $V_L$ chains of both Fabs are situated between the protrusions, with the $V_L$ chain of Fab1-1 being shifted further away from the 3-fold axis.

The two 5-fold Fabs differ in their binding behavior (Fig. 3c). While the $V_H$ chain of Fab1-6 is located close to the 5-fold channel, $V_H$ of Fab2-7 is located closer to the 2/5-fold wall. However, with its ten aa longer CDR3$_H$ loop, Fab2-7 is capable of interacting with the 5-fold channel. Furthermore, the $V_L$ chains of both Fabs interact with the AAV9 capsid in approximately the same region. Lastly, Fab1-2 binds to the 2/5-fold wall with its $V_H$ chain while the $V_L$ chain is situated between the 3-fold protrusions (Fig. 3d).

### The cryo-EM maps reveal the capsid-antibody contacts

The high resolution of the complex maps allowed the building of models for the Fabs and the capsid (Fig. 4a). Of particular interest were the CDRs of the Fabs that mediate the interaction with the capsid. The CDRs of the heavy chains (Fig. 4b) and light chains (Fig. 4c) exhibited well-resolved densities for the aa side chains, enabling the identification of the capsid-antibody contacts. Hydrogen bonds, van der Waals contacts, hydrophobic interactions, and salt bridges were considered as contacts (Fig. 4d). In some cases, the interactions involved intermediate densities, such as between N78 (Fab2-1) and T492, that were interpreted as water molecules. Other interactions involved bivalent cations, which were interpreted as calcium ions, e.g., between E125 (Fab2-3) and a series of AAV9 residues (Supplementary Fig. 2). All contacts for the Fabs from the three patients are listed in Supplementary Tables 2, 3, and 4. Overall, Fab1-3 had the lowest number of contacts with the AAV9 capsid, followed by Fab1-1. This coincided with them being among the Fabs with the smallest contact interface area. However, the Fab with the smallest interface area was Fab2-7 binding to the 5-fold region. The Fabs with the highest number of contacts and largest interface area all belong to group B of the 2-fold binders. Moreover, all Fabs interact with multiple VPs on the AAV9 capsid surface, which confirms that all mAbs exclusively detect intact capsids (Supplementary Fig. 3).

For the subsequent capsid engineering, the VRs and the frequency of the contacts on the capsid surface were analyzed. The epitopes for the sixteen 2-fold Fabs comprise VR-III to -VII, and -IX of the AAV9 capsid (Fig. 5a–c). However, only group B Fabs contact VR-III. On the aa level, all 2-fold-Fabs contact D532 and Y706. Other frequently contacted residues are T491, T492, R533, D556, N562, N704, and Y705, which are clustered around the 2-fold symmetry axis (Fig. 5d). The contact residues for Fab1-1 and Fab3-4 cluster around the 3-fold symmetry axis and involve aa in VR-V and -VIII, including T582 and Q588 (Supplementary Fig. 4). For the 5-fold binding Fabs, the contacted aa are more spread out and are located in VR-I, -II, -VII, the HI-loop, and VR-IX. Finally, the contacts of Fab1-2 are distributed to VR-I, -III, -VI, and VIII.

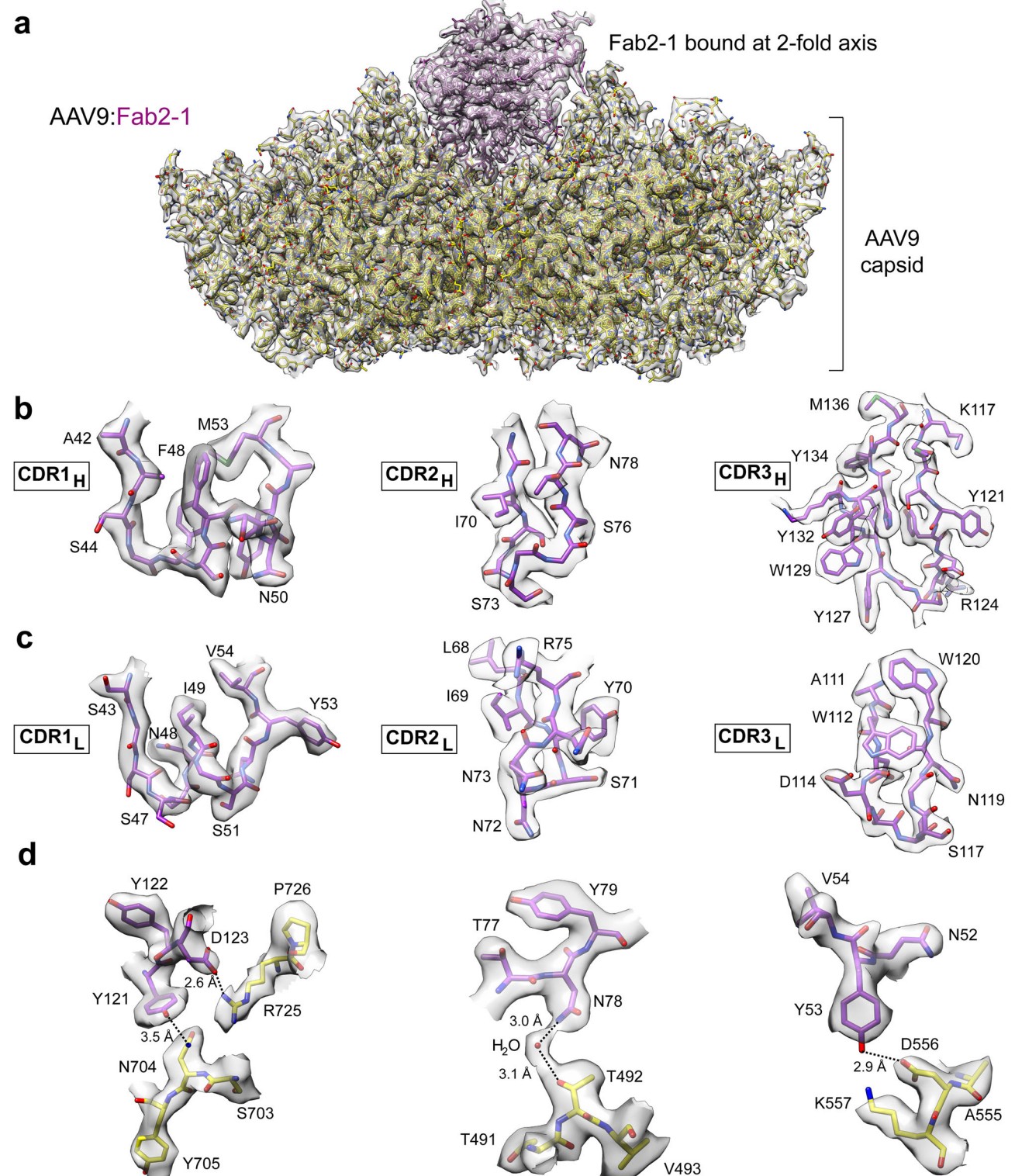

**Fig. 4 | Determination of the contacts of Fab2-1 to the AAV9 capsid. a** The fit of the AAV9 capsid (yellow) and Fab2-1 (purple) models inside the cryo-EM map is shown. For further detail, the modeled CDR1-3 of the **b** heavy chain and **c** light chain inside the cryo-EM map are displayed. **d** Exemplary interactions of the Fab to the AAV9 capsid are shown.

## The Fabs induce conformational changes of the AAV9 capsid

The high quality of the cryo-EM maps permitted the detection of induced conformational changes of the AAV9 capsid upon antibody binding. While the majority of the AAV9 capsid structure of the sub-particle maps was identical to previous AAV9 capsid structures, most Fabs induced structural changes at the Fab binding site. For Fab1-2,

binding to the 2/5-fold wall, the antibody caused a rearrangement of VR-I, shifting the position of some aa by up to 11 Å (Fig. 6a). Fab1-6 pushes the loop forming the 5-fold channel (VR-II) ~ 5 Å inwards restricting the diameter of the channel (Fig. 6b). The adjacent VR-II-loops not bound by the Fab adopt the canonical loop conformation of the AAV9 capsid in absence of the Fab. A similar observation was made

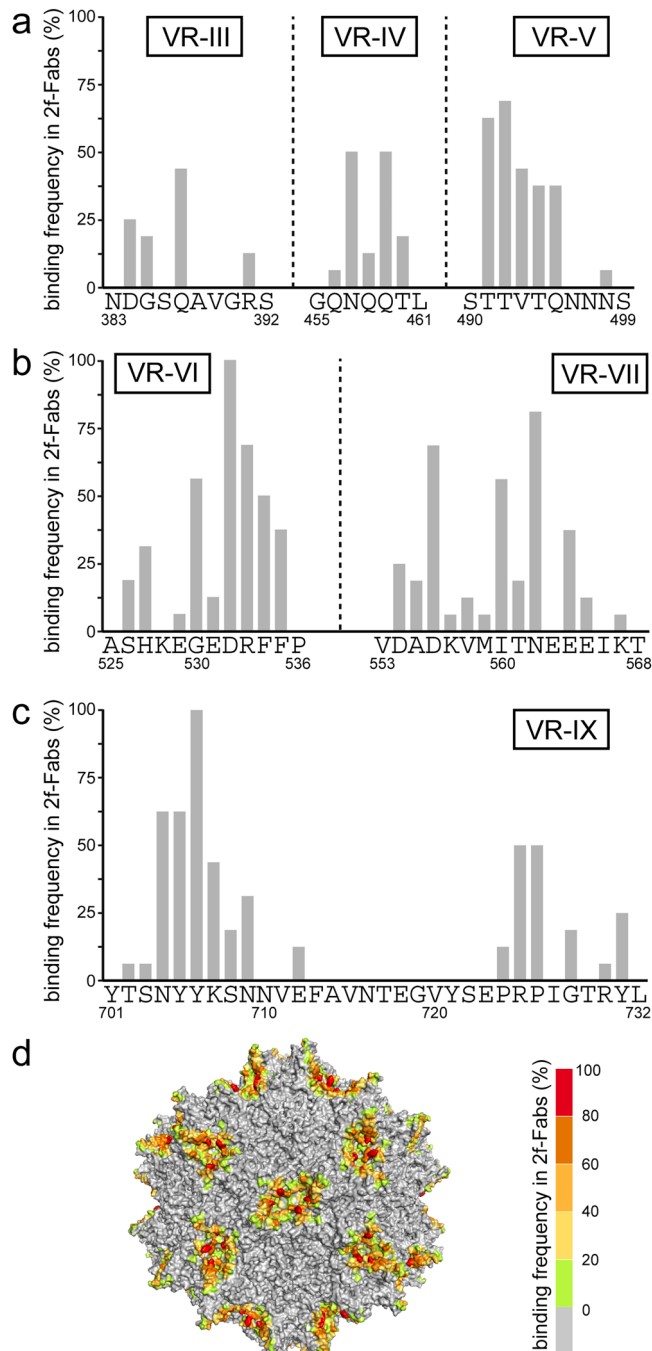

**Fig. 5 | Contact frequency of the 2-fold (2f) binding Fabs to the AAV9 capsid. a** The percentages of contacted capsid residues within variable region (VR)-III, -IV, -V, (**b**) -VI, -VII, and (**c**) VR-IX are shown for all 2-fold binding Fabs. **d** An AAV9 capsid surface representation with the binding frequency for any 2-fold binding Fabs is displayed which is colored according to the bar on the right.

for Fab1-1 where contact with Q588 altered the side chain orientation in one of the 3-fold protrusions (Fig. 6c), whereas the other two 3-fold protrusions adopt the unbound conformation. Fab1-7 and 3-3, belonging to group D of the 2-fold binders, interact with the apex of the 3-fold protrusions, resulting in shifts of VR-IV of up to 8 Å (Fig. 6d). Furthermore, for all the 2-fold Fab-complexes, movements occurred in VR-IX near the 2-fold axis. While, for AAV9 capsids in absence of Fabs the residues 704–707 adopt the same conformation in both symmetry-related VR-IX loops surrounding the 2-fold axis, there are numerous variations of side-chain orientations in the different Fab complex

structures (Supplementary Fig. 5). These changes are likely caused by the Fabs contacting these residues or by the Fab moving the capsid side chains into a more favorable position for antibody binding. Interestingly, none of the complex structures adopt the same conformation for residue range 704–707 in both symmetry-related VR-IX loops.

### Amino acid changes at the Fab contact sites enable antibody escape

The structural data of the AAV9-Fab complexes guided the development of AAV9 capsids with antibody escape phenotypes. Amino acid substitutions of the identified contact residues were selected either by disrupting critical interactions of the Fab with the capsid or by introducing steric clashes, preventing antibody binding (Supplementary Fig. 6). The capsids generated were analyzed for their productivity and infectivity to exclude non-viable variants. Subsequently, these variants were tested for their ability to escape the mAbs. Successful aa substitutions were combined to generate an AAV9 capsid variant with the fewest changes necessary, capable of evading multiple antibodies. The engineered capsids of this study are referred to as human antibody escape variants (hAEVs). These contain at least five aa changes, T491R, D556P, N562Y, T582Q, and Y706D (hAEV5) and non-density gradient-purified vectors show comparable empty-to-full capsid ratios compared to wtAAV9 (Supplementary Fig. 7). Additionally, variants with a sixth aa change, either Q588R or Q588Y, are called hAEV6 (Fig. 7a). In native immuno-dot blots hAEV5 was not recognized by mAb3-4 or any of the 2-fold-binding antibodies, amounting to 17 out of the 21 neutralizing human mAbs (Fig. 7b). The number of escaping antibodies can be further increased by changing Q588 to arginine or tyrosine, enabling the escape of mAb1-1, the other remaining 3-fold binding antibody (Supplementary Fig. 8). As none of the aa substitutions target the 5-fold and 2/5-fold wall binding mAbs, either hAEV was still detected by Fab1-2, Fab1-6, and Fab2-7. Several variants for these mAbs were developed, but either failed to escape the antibodies or resulted in defective capsids (Supplementary Table 5). When the transduction efficiencies of the capsid variants were compared to wtAAV9, hAEV5 was reduced by ~50%, hAEV6-Q588Y was near similar, and hAEV6-Q588R enhanced by ~8-fold (Fig. 7c). The introduction of the arginine suggested a gained receptor binding ability by the capsid and thus, a heparin competition assay was conducted. In this assay, the transduction efficiency of AAV9, hAEV5, and hAEV6-Q588Y was not affected by the presence of heparin, whereas hAEV6-Q588R's infectivity was reduced ~70% by heparin (Fig. 7d). This confirmed that hAEV6-Q588R likely gained the ability to utilize heparan sulfate proteoglycan as an attachment factor similar to AAV2[29].

When challenged with pooled neutralizing mAbs, that the hAEV capsids were designed to escape, hAEV5 and hAEV6-Q588R transduction were not inhibited, even in the presence of high mAbs concentrations (Fig. 7e). Instead, a slight enhancement of transduction was observed with increasing antibodies, similar to previous studies[20,21]. In contrast, AAV9 was nearly neutralized at a ratio of ~10 mAbs per capsid. Utilizing pooled mAbs, including all 21 mAbs did not alter the neutralization curve for AAV9 and also neutralized hAEV5 and hAEV6-Q588R at the highest to second-highest antibody concentrations. However, compared to AAV9, hAEV neutralization is shifted that ~10-100-fold higher mAbs concentrations are required for equivalent neutralization. This mimics the endpoint titer determined for human sera of six Zolgensma patients, which were ~9 and ~13-fold lower for hAEV5 and hAEV6-Q588R, when compared to the AAV9 capsid (Fig. 7f). In comparison to gene therapy recipients, antibody titers against the AAV capsids in healthy individuals are ~100-fold lower[25]. From a random set of fifty human sera from healthy donors, the endpoint titers of eight individuals with detectable antibodies against AAV9 decreased on average ~7-fold with the

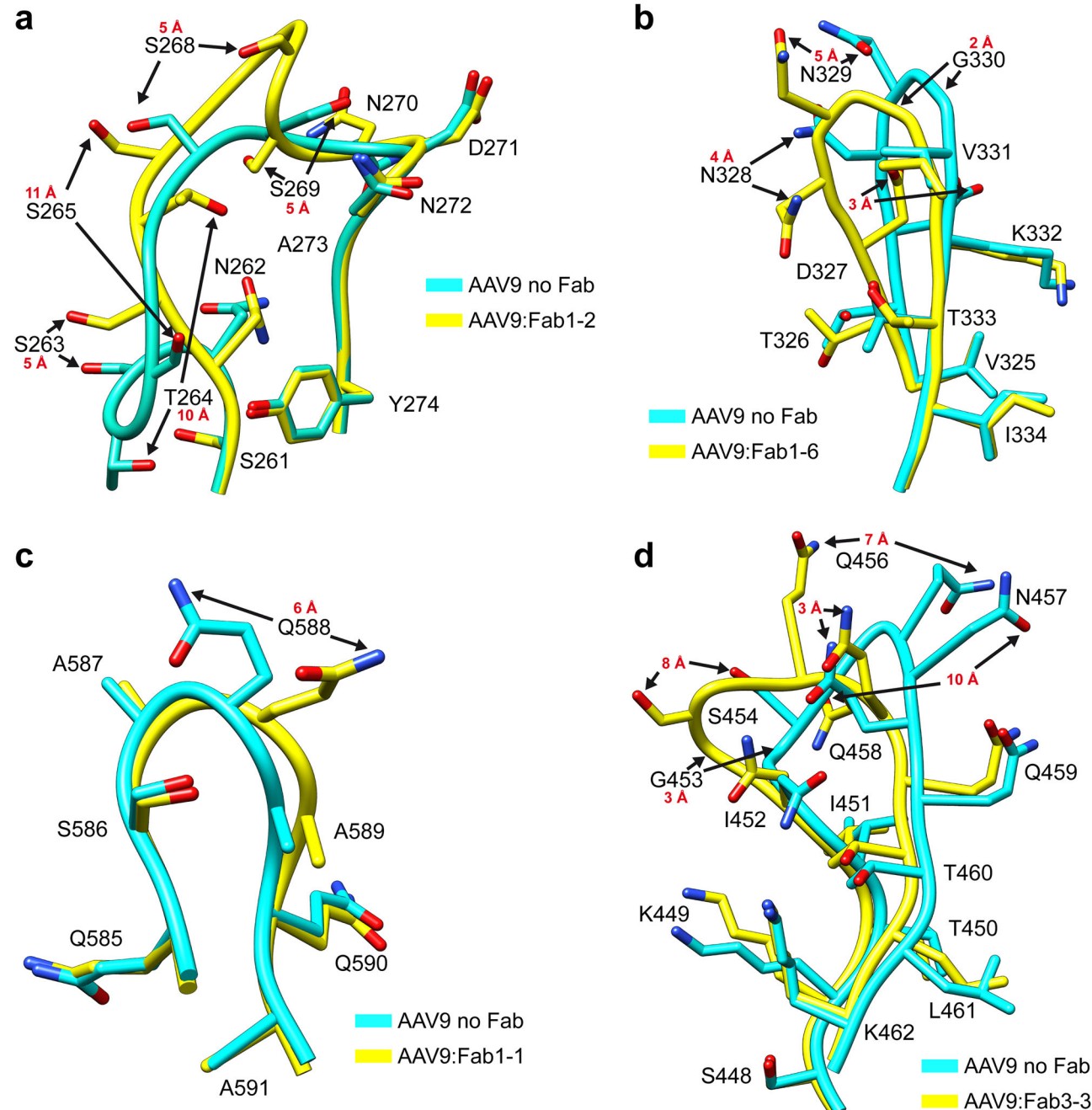

**Fig. 6 | Fab induced structural rearrangement of the AAV9 capsid. a** Binding of Fab1-2 to the AAV9 capsid leads to an alternative conformation of VR-I. The conformation in the absence of the antibody (cyan) and in the presence of Fab1-2 (yellow) are shown as ribbon diagrams with amino acid side chains. The amino acid residues are labeled and shown as stick representations with oxygen colored red and nitrogen blue. Movements of ≥3 Å measured in Coot between the atoms of the same amino acids for the two models are indicated. **b** Depiction as in (A) for VR-II upon binding of Fab1-6, **c** VR-IV upon binding of Fab3-3, and **d** VR-VIII upon binding of Fab1-1.

hAEV6-Q588R capsid (Supplementary Fig. 9a). Among the healthy donor sera not reactive against AAV9 capsid, only one serum showed minor reactivity against the engineered capsid. Similarly, the end-point titers of sera from children or mothers of infants precluded from AAV9-mediated gene therapies were reduced on average ~6-fold (Supplementary Fig. 9b). To show hAEV5's potential as an alternative to AAV9, its biodistribution was analyzed in C57BL/6 mice at a dose of $1.1 \times 1014$ vg/kg, which is the same dose administered to Zolgensma patients. While the transduction is reduced compared to AAV9 as described above (Fig. 7c and Supplementary Fig. 10), the biodistribution was largely unchanged (Fig. 7g).

## Discussion

To this date, all approved AAV biologics utilize the capsids of naturally occurring AAVs[30,31], which circulate in the human population, resulting in worldwide seroprevalence rates between 20-80% against different AAV serotypes[14,32]. While antibodies generated by the adaptive immune system are generally favorable to eliminate pathogens from the host and to prevent re-infections of the same agent, they are detrimental to the success of AAV-mediated gene therapy. To circumvent these antibodies, upcoming AAV biologics may need to utilize capsids where most, if not all, antigenic epitopes of the naturally circulating AAVs have been removed. These engineered capsids will not prevent

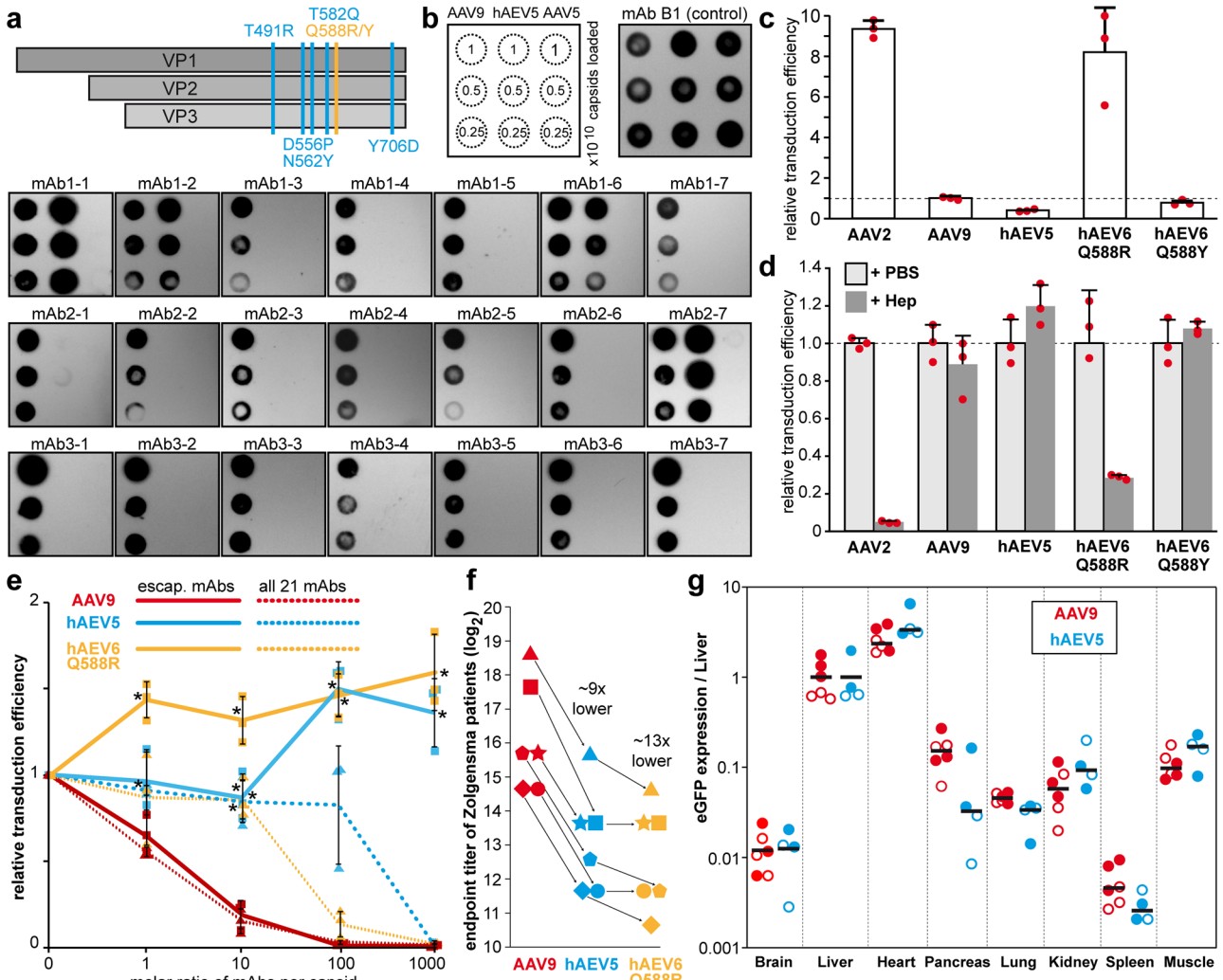

**Fig. 7 | Generation of an antibody escape variant. a** Schematic showing the amino acid substitutions of hAEV5 (blue) and hAEV6 (blue and orange). **b** Native dot blots are shown using mAb B1 as a loading control and the 21 human mAbs against the capsids of AAV9, hAEV5, and AAV5. The hAEV5 capsid variant escapes 17 of the 21 human antibodies, except for mAb1-1, 1-2, 1-6, and 2-7. **c** Transduction analysis of AAV2, AAV9, hAEV5, hAEV6-Q588R, and hAEV6-Q588Y vectors in HEK293 cells. The individual data points are indicated by a red circle. **d** Analysis of the transduction efficiency of the AAV vectors in absence (+PBS) and presence of 2 mg/ml heparin (+Hep). The efficiency is normalized to the transduction in absence of heparin. **e** Transduction of hAEV5 and hAEV6-Q588R is maintained in the presence of pooled mAb solutions, which contain equal amounts of the mAbs that the aa substitutions of the hAEV are targeted against (square data points) or equal amounts of all 21 mAbs (triangle data points). AAV9 vectors are analyzed in parallel. The *x*-axis represents the average molar ratio of mAbs added to the capsids. All experiments

have been performed in biological triplicate to verify reproducibility ($n = 3$), and the data (**c**–**e**) are presented as mean values ± standard deviation (SD). Statistical significance is calculated by a two-sided *t*-test for hAEV5 or hAEV6-Q588R compared to AAV9 at the same antibody condition. Asterisk indicates $p < 0.01$. **f** Sera from Zolgensma-treated infants ($n = 6$) were analyzed to determine the anti-AAV9, anti-hAEV5, and anti-hAEV6-Q588Y IgG endpoint titers. The individual patients (Pt) are represented by different shapes, and the average reduction of the endpoint titer from AAV9 to hAEV5 and hAEV6-Q588R is calculated. **g** Relative biodistribution compared to liver transduction of AAV9 and hAEV5 vectors in eight tissues following i.v. injection of $1.1 \times 10^{14}$ vg/kg in C57BL/6 mice. Fully colored and open-colored circles represent male mice and female mice. The average of the data ($n = 6$ for AAV9 and $n = 4$ for hAEV5) is indicated by the horizontal line. Source data are provided as a Source Data file.

future immune responses upon vector administration, but will expand the patient cohort treatable with AAV vectors that previously were precluded due to their pre-existing antibodies. In the past 13 years, the antibody responses against the AAV capsids were primarily simulated with mouse mAbs[13,21,33,34]. However, our recent study characterizing 21 anti-AAV9 capsid mAbs from human patients indicated that human and mouse antibodies may exhibit differential binding characteristics to the AAV capsids[25]. Prior studies showed that 66% of mouse mAbs bound to the 3-fold region, whereas 75% of human mAbs bound to the 2-fold region of the AAV9 capsid[20,33]. This observation was unusual because none of the 22 mouse mAb developed against AAV capsids to date have been observed to bind directly to the 2-fold region of any AAV capsid[13]. However, it should be noted that some antibodies, such

as 4E4, ADK4, and ADK9, bridge across the 2-fold axis without binding to the depressed region of the capsid[12,13,20]. Consequently, the 2-fold region had been considered to be the least antigenic region of the AAVs, potentially due to the Fab being sterically unable to enter the 2-fold depression. A possible explanation that allows human antibodies to contact the residues inside the depression is their longer CDRs compared to mouse antibodies, which were reported to be $15.5 \pm 3.2$ and $11.5 \pm 1.9$ aa for the human and mouse $CDR3_H$, respectively[35]. The lengths of the $CDR3_H$ loops of the 2-fold binding mAbs in the present study were even longer, ranging from 14-24 aa. In comparison, the $CDR3_H$ loop of mouse mAbs with known sequences binding to AAV capsids has a length of 12 (A20) and 11 aa (HL2476)[22,34]. However, the length alone does not explain the preference for the

2-fold region. A comparison of previously determined antibody-antigen interactions indicated that there is a tendency for hydrophobic aa to reside in the center of an antibody epitope, flanked by charged aa[36]. The same study also analyzed the aa preference within epitopes, with the top 4 overrepresented residues being tryptophan, tyrosine, methionine, and histidine, whereas valine, alanine, leucine, and cysteine were the most underrepresented residues. These conditions are met at the 2-fold region of the AAV9 capsid and may be the driver for the antigenic hotspot (Supplementary Fig. 11).

The HL2476-Fab bound to the capsid of AAV5 is currently the only capsid-antibody complex determined at high-resolution (3.1 Å)[22]. This Fab was found to bind to the 3-fold protrusion, a total of 60 times. As such, the Fab followed the icosahedral symmetry of the capsid, enabling the modeling of the complex. The remaining AAV-Fab complexes have been determined at resolutions ranging from 4.1 to 23 Å[13,20,21,37]. If high-resolution structure determinations using cryo-EM and conventional icosahedral reconstruction would have been attempted for antibodies binding near the 2-fold (e.g., 4E4, ADK4, ADK9), at the 3-fold (e.g., 5H7, PAV9.1), or around the 5-fold (e.g., HL2372) the structural information for the Fabs would have been lost due to icosahedral averaging of the capsid, similarly as observed in this study. Binding of Fabs directly at one of the symmetry axes precludes another Fab from binding in one of the other symmetry-related binding modes. As a result, Fabs binding at the 2-, 3-, and 5-fold axis are limited to 30, 20, and 12 binding events to the icosahedral capsid, respectively. The 5-fold binding Fabs in this study bind offset from the symmetry axis. However, due to the fact that Fab1-6 would be clashing with Fabs bound to its symmetry-related neighboring binding sites, this Fab is limited to 24 binding events to the icosahedral capsid. In contrast, no clashes were observed for Fab2-7, enabling the binding of up to 60 Fabs to the capsid. To resolve symmetry mismatches, localized reconstruction methods have been developed to structurally characterize patches of the capsid for ligands not conforming to the icosahedral symmetry[26–28]. Prior to this study, asymmetric features of AAV capsids have not been analyzed, except for a study of AAV2 in complex with AAVR determined by cryo-electron tomography at 20–30 Å resolution and a recent study that used a similar strategy to localized reconstruction for an AAV9 capsid variant binding to its receptor carbonic anhydrase IV[38,39].

For some of the capsid-Fab complexes, ordered water molecules were observed in the binding interface. This has been previously observed for antigen-antibody structures and suggested to mediate and stabilize the antigen-antibody association[40]. Additionally, at least two Fabs showed the presence of a bivalent cation in the binding interface. Previous studies have shown that some antibodies depend on the presence of calcium ions for the binding of its antigen[41,42]. Future characterizations of the mAbs will confirm if capsids treated with EDTA will prevent antibody binding or reduce their binding affinity.

Conformational changes in the antibody following antigen binding, especially in the CDRs, have been widely reported[19,43–45]; however, studies observing structural changes in the antigen upon antibody binding, as observed here, are less common. However, a recent study showed that binding of antibodies to SARS-CoV-2 variants induced conformational changes in the viral spike protein[46]. These observations are only detectable at sufficient resolution. For the AAV5-HL2476 complex, only minor shifts up to 1.1 Å in VR-V were observed, which was the main contact region of the Fab[22]. Similarly, conformational changes in the capsid of a distantly related parvovirus (canine parvovirus), upon binding with Fab14, were also minor, with a maximum displacement of the main chain by 1.9 Å[47]. For the Fabs in this study conformational changes of the main chain varied from minor (Cα-RMSD ≤ 2 Å) for the 3-fold and most 2-fold binding Fab; medium changes (Cα-RMSD 2–5 Å) for the 5-fold Fabs; to significant remodeling of the surface loops (Cα-RMSD 5–11 Å) for the group D 2-fold Fabs and

Fab1-2 binding to the 2/5-fold wall. Additionally, many aa side chains in the epitopes also adopt alternative rotamer conformations. This was most pronounced for the 2-fold binding Fabs, leading to various conformations of aa 704-NYYK-707. The 2-fold region of the AAV capsids has been suggested to play a role in viral transcription[48] and may require similar dynamic aa configurations for this function.

Side chain orientation shifts can be caused by interactions with the Fab, making these residues prime targets for capsid engineering, exemplified by the Q588R variant, which escapes Fab1-1. Previous AAV9 variants developed for the escape of PAV9.1 swapped the residues aa586-590 with the corresponding residues from other AAV serotypes[33] that may also escape Fab1-1. However, to prevent the reintroduction of antigenic epitopes from other AAVs, a more targeted approach was conducted, facilitated by the higher resolution structure, and Q588 was substituted to a residue not found in other AAV serotypes. The AAV9 variants developed to escape the other mouse mAbs, S454A and P659K[20], had no impact on the human mAbs, but vice versa, Q588R also allowed escape from ADK9. The hAEV5 and hAEV6-Q588R capsids were able to prevent neutralization from 17 or 18 of the 21 human mAbs, respectively, comprising all AAV9-specific antibodies[25]. To date, no single aa substitution capsid variant has been identified capable of escaping the mAbs binding to the 5-fold region and the 2/5-fold wall, which cross-react to multiple AAV serotypes, including non-primate AAVs[49]. Likely, multiple aa substitutions will be required to disrupt the binding of these antibodies.

When challenged with a mix of all neutralizing mAbs, simulating a polyclonal antibody response, the current hAEV capsids tolerate ~10-fold higher antibody concentrations before being neutralized compared to wtAAV9. This mirrors the observed reduction of endpoint titers against the hAEV capsid for sera of Zolgensma patients and healthy donors reactive against AAV9 capsids. While anti-AAV9 antibody titers in patients following SMA gene therapy are highly elevated, the titers of individuals with naturally acquired anti-AAV9 antibodies are ~100-fold lower[25]. Currently, the exclusion criteria for receiving Zolgensma are anti-AAV9 antibody titers >1:50. A recent serology study in the US concluded that ~13% of the tested children should not receive Zolgensma due to potential safety and efficacy issues[50]. Assuming a similar ~9-fold reduction of seroreactivity, the hAEV5 capsid would allow 64 of the precluded 115 children (~56%) to qualify for treatment.

Future studies will need to increase the transduction efficiency and specificity of the utilized AAV vectors. Notably, while the biodistribution of hAEV5 capsid is unchanged when compared to AAV9, liver transduction of both capsids is ~100-fold more efficient. The introduction of arginine in the hAEV6-Q588R capsid increases the transduction efficiency due to its position being in a similar position to the arginines mediating heparin binding in the AAV2 capsid[51]. However, heparin-binding by AAV capsids was previously found to negatively impact CNS transduction[52], thus likely altering the biodistribution. Recently, several engineered capsids with peptide library insertions in VR-VIII to improve CNS transduction have been developed[53,54]. These peptide insertions are often inserted between residues 588 and 589 and would guarantee the escape from mAb1-1, similar to Q588R or Q588Y. Thus, the combination of the hAEV5 capsid with the peptides has the potential to be a superior alternative for AAV9 vectors and will lead to a significantly lower rejection rate of patients in need of this life-saving medication, and possibly also allows redosing of patients previously treated with AAV9 vectors.

## Methods

The research conducted in this study complied with ethical regulations as defined by both the National Institutes of Health (NIH) and the University of Florida. Human ethics approval was granted (LNR/18/SCHN/522) by the Ethics Committee of the Sydney Children's Hospital Network for the collection of analysis blood from patients and healthy controls. Samples were collected from pediatric patients and healthy

adult controls after guardians and subjects, respectively, gave informed consent. Patient's data with respect to age and gender was de-identified to protect the privacy of the small number of individuals in the study. All animal care and experimental procedures were evaluated and approved (C402) by the CMRI/CHW Animal Care and Ethics Committee of the Children's Medical Research Institute and Children's Hospital at Westmead.

## Cell culture
HEK293 cells (ATCC, cat#: CRL-1573) were maintained adherent in Dulbecco's Modified Eagle Medium (DMEM) (Thermo Fisher, Waltham, MA), supplemented with 10% heat-inactivated fetal calf serum and 100 units of penicillin/ml and 100 µg of streptomycin (Caisson Laboratories, Smithfield, UT) at 37 °C in 5% $CO_2$.

## Site-directed mutagenesis
The R2V9 plasmids containing AAV2 *rep* and AAV9 *cap* gene served as the template for site-directed mutagenesis PCR reactions. For each mutant, complementary PCR primers were designed that contained the desired mutation (Supplementary Table 6), which was flanked on both sides by 10 to 15 homologous base pairs. Primers were ordered from Sigma-Aldrich (Houston, TX) and used in PCR amplification reactions using a C1000 Touch™ thermal cycler (Bio-Rad, Hercules, CA) and *Pfu* Ultra high-fidelity DNA polymerase (Agilent, Santa Clara, CA). PCR products were incubated at 37 °C for 1 h with *Dpn*I restriction enzyme (NEB, Ipswich, MA) to degrade the methylated template plasmid. The reactions were then transformed into Top10 competent cells (Thermo Fisher), which were cultured on LB-ampicillin selective media and further amplified to isolate the plasmid. Clones were submitted for Sanger sequencing (Genewiz, South Plainfield, NJ) to verify the introduced mutations.

## Recombinant AAV production and purification
Recombinant AAV vectors with a packaged luciferase gene were produced by triple transfection of HEK293 cells at a confluency of ~80–90%, utilizing pTR-UF3-Luciferase, pHelper (Stratagene), and R2V9[55] or variants thereof. The transfected cells were harvested 72 h post-transfection, washed with phosphate-buffered saline (PBS; 137 mM NaCl, 2.7 mM KCl, 100 mM $Na_2HPO_4$, 2 mM $KH_2PO_4$), the cells pelleted and resuspended in PBS with 1 mM $MgCl_2$ and 2.5 mM KCl. The resuspended cells were subjected to three freeze-thaw cycles (−80 °C to 37 °C) and subsequently incubated with 125 units/mL benzonase for 1 h at 37 °C before centrifugation at $10,000 \times g$ for 15 min to pellet the cell debris. AAV9 vectors or variants thereof were purified by Capture Select AAV9 affinity chromatography[56]. The purity of the AAV capsid samples was analyzed by SDS-PAGE. For the determination of the titer of vector genome-containing capsids, 5 µL aliquots were incubated with proteinase K (Sigma-Aldrich) at 56 °C for 2 h, and the released vector genomes were purified by a PCR purification kit (Invitrogen). Subsequently, a quantitative PCR was conducted in a Bio-Rad MyiQ2 Thermocycler instrument (Bio-Rad) using primers amplifying a 146 bp segment of the luciferase gene (Supplementary Table 6) and the SYBR Green Master Mix (Bio-Rad).

## Native dot immunoblot analysis
Native AAV9 capsids were adsorbed onto nitrocellulose membranes (Bio-Rad, Hercules, CA) in a dot blot manifold (Schleicher and Schuell, Dassel, Germany). For loading controls and to confirm whether the capsids are exclusively detected in their native state, AAV9 capsids were heated at 100 °C for 5 min prior to adsorption to the membrane. Excess fluid was drawn through the membrane by vacuum filtration. The membrane was removed from the manifold and blocked with 6% milk in PBS, pH 7.4, for 1 h. The human mAbs (produced in prior study)[25] or mAb B1 (ARP, catalog # 690058) were applied to the membrane at a concentration of ~1 µg/ml in PBS with 6% milk, 0.1%

Tween-20, and incubated for 1 hr. The membrane was then washed with PBS, and an anti-human-HRP (Abcam, cat# ab6858) or anti-mouse-HRP secondary antibody (Cytiva cat# NA931-1ML) was applied at a dilution of 1:50000 or 1:3000 in PBS/6% milk, respectively, and incubated for 1 h. The membrane was washed with PBS, and then Immobilon™ Chemiluminescent Substrate (Millipore, Darmstadt, Germany) was applied to the membrane, and the signal was detected on X-ray film.

## AAV transduction, neutralization, and heparin competition assay
Purified AAV9-luciferase vectors (or variants) were used to infect HEK293 cells seeded on 24 well-plates at an MOI (multiplicity of infection) of 100,000. After 48 h, cells were lysed and luciferase activity assayed using a luciferase assay kit (Promega, Madison, WI) as described in the manufacturer's protocol. Uninfected cells of the same plate were used as negative control. For neutralization assays, the vectors were pre-incubated for 30 min at 37 °C with either purified mAbs or PBS as the negative control at variable ratios relative to the capsids prior to infection. The transduction efficiency was calculated as a percentage to AAV vectors in the absence of antibodies. For the heparin competition assay, heparin (Sigma) was pre-incubated with the AAV9-luciferase vectors in a volume of 50 µL at a concentration of 2 mg/ml for 30 min at 37 °C. The old media of the cells in the 24 well-plate was removed and replaced with the AAV-heparin mixture. Following an incubation for 90 min the mixture was removed and fresh DMEM medium added to the cells. The luciferase expression was analyzed after 48 h.

## Cryo-EM sample preparation and data collection
Purified AAV9 vectors were mixed with Fabs at a ratio of ~2 Fabs per potential VP binding site in the capsid, resulting in a final ratio of ~1:120 (capsid to Fab)[25]. Three and a half microliters of the AAV9-Fab complex samples were applied to glow-discharged Quantifoil copper grids with 2 nm continuous carbon support over holes (Electron Microscopy Sciences), blotted for 5 s (blot force: 5), and vitrified in liquid ethane using a Vitrobot Mark 4 (FEI) at 95% humidity and 4 °C. The grids were screened for particle distribution and ice quality with an FEI Tecnai G2 F20-TWIN microscope (FEI Co., Hillsboro, OR, USA) operated under low-dose conditions (200 kV, ~20 e⁻/Å²). Images were collected on a Gatan UltraScan 4000 CCD camera (Gatan, Inc., Pleasanton, CA, USA). High-resolution data for the AAV9-Fab complexes were collected at the Stanford-SLAC Cryo-EM Center ($S^2C^2$), using a Titan Krios (FEI) electron microscope operated at 300 kV, equipped with a Falcon 4 direct electron detector (Thermo Fisher). A total of 50 movie frames were collected per micrograph at a total electron dose of ~50 e⁻/Å², a defocus range of −0.8 to −2.1 µm, and a pixel size of ~0.72, 0.83, or 0.93 Å/pixel. Movie frame alignment was conducted using MotionCor2 (version 1.5) with dose weighting[57].

## 3D reconstruction of the AAV9: Fab complexes
For the initial reconstruction using icosahedral averaging cisTEM (version 1.0.0) was utilized (Supplementary Fig. 12)[58,59]. Briefly, the motion-corrected micrographs were imported into the program, and their contrast transfer functions (CTFs) were calculated. Micrographs of poor quality were removed. The capsid-Fab complexes were automatically picked using a characteristic particle radius of 135 Å, and the individual capsid images extracted. These were then sorted via 2D classification into 50 classes. Classes containing impurities were discarded. The ab initio 3D function was utilized to generate an initial map using 10% of the particles. This map was further refined using the automatic refinement function with default settings. The final electron density maps with imposed icosahedral averaging were sharpened using the pre-cut off B-factor value of −90 Å² and variable post-cut off B-factor values of 0, 20, and 50 Å².

In order to conduct localized reconstructions, the stack of individual particle images and the parameter file were exported in cisTEM to the Relion format (Supplementary Fig. 13). These particles were imported in Scipion 3 with the pwem protocol and all further steps conducted in this program. As the first step, the subparticles were defined by specifying the vector length from the center of the particle either along the 2-fold, 3-fold, or 5-fold symmetry axis (protocol: localrec−define subparticles). In the subsequent step, the subparticles are filtered, keeping only unique particle images and removing view angles that significantly overlap with capsid (protocol: localrec−define subparticles). The individual images of the remaining subparticles are extracted (protocol: localrec−extract subparticles) and used for the reconstruction of an initial, low-resolution map with a C1 symmetry operator (protocol: relion−reconstruct). This map was then used as an input volume for the 3D classification protocol, asking for at least four selected classes. During this step, C2, C3, or C5 symmetry relaxation is activated. The resulting maps of the individual classes were inspected, and the map with the highest resolution features for the capsid-Fab complex, as well as the particles contributing to "good" classes selected for further refinement using the relion−3D auto-refine protocol. The final map was sharpened using the relion−post-processing protocol. The resolutions of all reported maps were determined at a Fourier shell correlation (FSC) criterion threshold of 0.143 (Supplementary Figs. 14−17). The local resolution was estimated with CryoSPARC utilizing the half maps of the prior 3D reconstruction[60].

### Docking of the AAV9 capsid and Fab models and scaling of the cryo-EM map

The atomic coordinates of the 60-meric AAV9 capsid (PDB accession number 3UX1) were docked into the icosahedral-averaged cryo-reconstructed density maps using the 'fit in map' subroutine in UCSF-Chimera[61]. In order to identify the true pixel size of the maps, a series of correlation coefficient (CC) calculations with different voxel sizes were conducted. The pixel size with the highest CC was utilized to resize the reconstructed map with the e2proc3d.py subroutine in EMAN2[62]. Subsequently, the map was normalized and converted to the CCP4 format using MAPMAN (version 7.8.5)[63]. Structure models of the heavy and light chains of the Fabs were conducted with RoseTTAFold using the primary amino acid sequences[64]. The resulting PDB files were docked into the icosahedral-averaged or localized reconstructed CCP4 maps by rigid body rotations and translations and by using the 'Fit-in-map' function in Chimera.

### Model refinement

The AAV9 capsid and the Fab light and heavy chain coordinates were refined in Coot (version: 0.8.9.2)[65] by manual building and utilizing the real-space-refinement subroutine to adjust side- and main-chains into the CCP4 map. The complexes were further refined against the maps using the real-space-refine subroutine in Phenix (version: 1.10.2155) with default settings, which also provided refinement statistics (Supplementary Table 1)[66]. For the graphical representations of the maps and models, Chimera and PyMol were utilized[61,67]. The contact interactions for the AAV9-Fab complexes were analyzed using the online server PDBePISA[68].

### Enzyme-linked immunosorbent assay

Enzyme-linked immunosorbent assays (ELISA) were performed to determine the antibody endpoint titer for human sera against AAV9 or the engineered capsids[25]. Samples were collected from pediatric patients and healthy adult controls after guardians and subjects, respectively, gave informed consent. The patient's data with respect to age and gender was de-identified to protect the privacy of the small number of individuals in the study. AAV vector preparations (50 μL) at a concentration of $2.5 \times 10^{10}$ capsids/mL (in coating buffer (carbonate-bicarbonate buffer, Sigma-Aldrich) were used to coat 96-well

polystyrene Maxisorp ELISA plates (Nunc) overnight at 4 °C. The plates were washed 3 times with PBS + 0.05% Tween-20 (Sigma-Aldrich) and then received 100 μL per well of blocking buffer (PBS + 5% skim milk + 0.05% Tween-20). After incubation at room temperature (RT) for 2 h the plates were washed 3 times with PBS and received 50 μL per well of sera (serially diluted in blocking buffer as indicated, with duplicate wells for each dilution). The plates were incubated for 2 h at RT and washed 3 times with PBS before receiving 50 μL per well of Goat Anti-Human-heavy IgG chain-HRP conjugated antibody (AP309P, Sigma-Aldrich) diluted 1:10,000 in blocking buffer. Plates were incubated for 1 h at RT and washed 4 times before receiving 75 μL per well of 3,3′,5,5′-Tetramethyl-benzidine (TMB, Sigma-Aldrich). Plates were incubated in the dark for 30 min at RT before reactions were stopped using 75 μL per well of 1 M $H_2SO_4$. The absorbance of each well was measured at 450 nm wavelength using a VersaMax microplate reader (Molecular Devices, LLC). Duplicate wells containing no AAV served as background controls for each sera dilution. The mean value for each sample dilution was calculated for wells both with (foreground) and without coated vector (background), and the endpoint titer was determined as the lowest dilution where this ratio is >8.2. The limit of sensitivity for each assay is indicated in the graphs. Human ethics approval (LNR/18/SCHN/522) permitted collection and assay of sera from SMA patients after treatment with Zolgensma.

### Determination of biodistribution

For the biodistribution study, a GFP expression cassette was packaged as in AAV9 or hAEV5 capsids. Vectors were purified by ultra-centrifugation through cesium chloride gradients[69] and titered by qPCR[70]. Male and female C57BL/6 mice (*Mus musculus*) received AAV vector ($1.1 \times 10^{14}$ vg/kg) in 150 μL PBS via tail vein injection at 8 weeks of age. Mice were euthanized via cardiac puncture whilst under anesthetic (oxygen/fluorane) without waking, followed by cervical dislocation. Tissues were recovered 3 weeks post-injection, snap frozen in liquid nitrogen, and stored at −80 °C until analysis. For analysis of GFP expression, 0.01–0.02 grams of tissue was homogenized in 300 μL lysis buffer, mixed at 4 °C for 1 h, and subjected to centrifugation ($800 \times g$) at 4 °C for 20 min. Supernatants were assayed for protein concentration using a DC Protein Assay kit (Bio-Rad) as per the manufacturer's instructions. The GFP concentration in lysates was determined by fluorometry at excitation and emission wavelengths of 485 nm and 535 nm, respectively, and was calculated from a standard curve. The gastrocnemius was analyzed as a source of muscle. All animal care and experimental procedures were evaluated and approved by the Animal Care and Ethics Committee of the Children's Medical Research Institute and Children's Hospital at Westmead. Male and female C57Bl/6 mice were purchased from the Australian Bioresources (Moss Vale, Australia). Mice were housed in vented boxes held at 19–21 °C and 45–65% humidity and were subjected to 12-h light/dark cycles. Mice received normal food and water ad libitum for the duration of the experiments.

### Reporting summary

Further information on research design is available in the Nature Portfolio Reporting Summary linked to this article.

## Data availability

The generated cryo-EM reconstructed density maps in this study have been deposited in the Electron Microscopy Data Bank (EMDB) under the accession codes EMD 44271 (AAV9-Fab1-1), EMD 44272 (AAV9-Fab1-2), EMD 44273 (AAV9-Fab1-3), EMD-44274 (AAV9-Fab1-4), EMD-44275 (AAV9-Fab1-5), EMD-44276 (AAV9-Fab1-6), EMD-44277 (AAV9-Fab1-7), EMD-44314 (AAV9-Fab2-1), EMD-44315 (AAV9-Fab2-2), EMD-44316 (AAV9-Fab2-3), EMD-44317 (AAV9-Fab2-4), EMD-44318 (AAV9-Fab2-5), EMD-44319 (AAV9-Fab2-6), EMD-44320 (AAV9-Fab2-7), EMD-44321 (AAV9-Fab3-1), EMD-44322 (AAV9-Fab3-2), EMD-44323 (AAV9-Fab3-3),

EMD-44324 (AAV9-Fab3-4), EMD-44325 (AAV9-Fab3-5), EMD-44326 (AAV9-Fab3-6), and EMD-44327 (AAV9-Fab3-7). The atomic models built in this study have been deposited in the Protein Data Bank (PDB) under the accession numbers 9B6N (AAV9-Fab1-1), 9B6O (AAV9-Fab1-2), 9B6P (AAV9-Fab1-3), 9B6Q (AAV9-Fab1-4), 9B6R (AAV9-Fab1-5), 9B6S (AAV9-Fab1-6), 9B6T (AAV9-Fab1-7), 9B7K (AAV9-Fab2-1), 9B7L (AAV9-Fab2-2), 9B7M (AAV9-Fab2-3), 9B7N (AAV9-Fab2-4), 9B7O (AAV9-Fab2-5), 9B7P (AAV9-Fab2-6), 9B7Q (AAV9-Fab2-7), 9B7R (AAV9-Fab3-1), 9B7S (AAV9-Fab3-2), 9B7T (AAV9-Fab3-3), 9B7U (AAV9-Fab3-4), 9B7V (AAV9-Fab3-5), 9B7W (AAV9-Fab3-6), and 9B7X (AAV9-Fab3-7). A PDB code of the previously published AAV9 capsid structures used in this study is 3UX1. Source data are provided as a Source Data file. Source data are provided with this paper.

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

## Acknowledgements

The authors would like to thank the UF-ICBR electron microscopy core for access to electron microscopes utilized for cryo-electron micrograph screening (RRID:SCR_019146). High-resolution cryo-EM data collection was performed at the Stanford-SLAC Cryo-EM Center (S$^2$C$^2$), which is supported by the National Institutes of Health Common Fund Transformative High-Resolution Cryo-Electron Microscopy program (U24 GM129541). The content is solely the responsibility of the authors and does not necessarily represent the official views of the National Institutes of Health. The study was funded by an NIH grant R01 NIH GM082946 (to RM), a National Health and Medical Research Council of Australia grant (APP2004320 and APP2029992 to IEA and GJL; and APP1194940 to MAF), the Rebecca Cooper Foundation (PG2019449 to GJL) and a Research Council of Finland grant (348021 to JTH).

## Author contributions

M.M. and R.M. conceived the project. J.H., A.R.N., N.K., and A-M.H. produced the AAV vector samples used for structure determination, in vitro and in vivo studies. P.C. vitrified the sample grids and screened them prior to high-resolution data collection. M.M., M.G., and J.T.H. conducted cryo-EM data processing and 3D reconstructions. M.M. built the atomic models, analyzed, and validated the cryo-EM structures. M.M., J.H., A.R.N., J.Z., and L.P. designed, created, and tested the AAV9 capsid variants. N.K., A-M.H., and N.JC.S. conducted the biodistribution and serology studies. M.A.F. coordinated ethics approval, recruitment, consent, biospecimen collection, and processing for the isolation of monoclonal antibody and serology data used in this study. M.M., I.E.J., G.J.L., J.T.H., and R.M. provided supervision and acquired funding. M.M. wrote the original draft of the manuscript, and all authors provided inputs and edits.

## Competing interests

IAE and GJL are in advanced discussion with a biotechnology company regarding the commercial manufacture and distribution of the monoclonal antibodies (mAbs) described in this manuscript. These negotiations may result in financial compensation for IAE and GJL and their affiliated organizations. JTH is co-founder and CEO of Nanometria, a limited liability company. The University of Florida Research Foundation, Inc. has filed a patent application (PCT/US2024/015006 [https://patents.google.com/patent/WO2024168153A2/en?oq=PCT%2fUS2024%2f015006]) on behalf of MM, JH, ARN, and RM based on the capsid variants described in this study. The remaining authors declare no competing interests.

## Additional information

[1]Department of Biochemistry & Molecular Biology, Center for Structural Biology, McKnight Brain Institute. College of Medicine, University of Florida, Gainesville, FL, USA. [2]Gene Therapy Research Unit, Children's Medical Research Institute, Faculty of Medicine and Health, The University of Sydney and Sydney Children's Hospitals Network, Westmead, NSW, Australia. [3]Discipline of Paediatrics, University of Adelaide, Women's and Children's Hospital, North Adelaide, SA, Australia. [4]Department of Neurology and Clinical Neurophysiology, Women's and Children's Health Network, North Adelaide, SA, Australia. [5]School of Clinical Medicine, UNSW Medicine and Health, UNSW Medicine, Sydney, NSW, Australia. [6]Department of Neurology, Sydney Children's Hospital, Randwick, NSW, Australia. [7]Interdisciplinary Center of Biotechnology Research, University of Florida, Gainesville, FL, USA. [8]Institute of Biotechnology, Helsinki Institute of Life Science HiLIFE, University of Helsinki, Helsinki, Finland. [9]Discipline of Child and Adolescent Health, University of Sydney, Westmead, NSW, Australia. ✉e-mail: mario.mietzsch@ufl.edu; rmckenna@ufl.edu

