## [Transparent Peer Review file · Nature Communications]

Structural characterization of antibody-responses following Zolgensma treatment for AAV capsid engineering to expand patient cohorts

Corresponding Author: Dr Mario Mietzsch

Version 0:

Reviewer comments:

Reviewer #1

(Remarks to the Author)

The manuscript Structural characterization of antibody-responses from Zolgensma treatment provides a blueprint for the engineering of an AAV capsid suitable for redosing submitted by Mietzsch and colleagues provides a unique insight on how human antibodies bind to AAV9 capsids.

The monoclonal antibodies assayed within this study are derived from patients receiving Zolgensma. Thus, extremely high vector doses (10E14 per kilogram body weight) had been applied to presumably very young patients which both might have an impact on the antibody pool present in the patients following gene therapy. The possible impact of those parameters should be discussed by the authors. Moreover, is the source - like in the previous publication - three infants?

According to the knowledge of the reviewer, six and not seven AAV gene therapies have received market approval. The authors might edit the introduction accordingly.

The AAV capsid is composed of 60 subunits to which three VPs contribute. This is different from the authors' statement on 60 copies of the viral proteins. The authors need to revise their statement.

One caveat of this very interesting study is that it remains unclear which of the antibodies are neutralizing antibodies. This is of high interest as authors report on differences in the structures recognized by murine compared to human antibodies AND on conformational changes of the capsid upon antibody binding. This information must be included.

The authors cloned a new capsid with six aa changes (T491R, D556P, N562Y, T582Q, Y706D). Why were these changes and not additional or other residues identified in their study used? On which bases were the aa substitution chosen. When describing Fab binding aa T584 is mentioned. The new variant contains a T582Q substitution. Is there a mistake in the numbering?

The authors did not include aa substitutions for residues at the 5-fold and 2/5-fold wall because they are non-infections. However, Table S5 is lacking details on how infectivity was defined.

In the presence of antibodies, hAEV6 shows an increased transduction efficiency. How is this explained?

hAEV6 and AAV9 differ in infectivity. How do the authors explain the higher infectivity of hAEV6 and was this difference considered when performing the neutralization assay? This is crucial to state that the hAEV6 is indeed an immune escape variant.

On the same line, authors state in the final paragraph that >While none of the changes in the hAEV6 variant targeted these antibodies, surprisingly, a shift in the neutralization assay was observed, allowing this capsid variant to tolerate ~10-fold higher antibody concentrations before being neutralized. This effect could be associated with the higher transduction efficiency of the hAEV6 variant.>

If this is indeed the case, the question on the value of the antibody binding study for the reported capsid engineering arises.

Moreover, is there a difference in empty-to-full capsid ratio between hAEV6 and AAV9? This is important since antibodies

bind the capsid independent on whether they contain a vector genome or not.

Since multiple changes to the capsid were made to generate hAEV6, a biodistribution study needs to be performed. Infectivity data are so far limited to a single cell line, which is highly permissive for several serotypes. Since authors state that hARV6 with its improved infectivity might be used as an alternative to AAV9, the mentioned biodistribution study is required.

Reviewer #2

(Remarks to the Author)

Mietzsch et al. characterize the structure of several human-derived monoclonal Abs with AAV9 capsids, suggesting that many of them bind by inducing conformational changes. Mutagenesis performed with the AAV9 capsid to disrupt Ab epitopes indicates the path towards new capsid variants that may escape pre-existing or treatment-emergent humoral immune responses.

I am not able to comment on the validity of processing structural data using localized reconstructions and symmetry relaxation and will defer to other reviewers who are experts in this field. The empirical basis for induced conformational changes should also be carefully evaluated by these structural experts, since I am unclear what the quantitative threshold for such change might be, as compared to simple "noise" in the cryo-EM data generated for the Ab-capsid complexes, or whether the observed conformational changes were an artefact of the localized reconstruction models.

Nonetheless, the derived models for the complexes predicted that individual amino acids on the capsid surface are involved in Ab binding, which was tested by targeted mutagenesis. The newly generated AAV9 capsid variant (hAEV6) carries 6 aa changes and eliminates binding to 18 of the 21 monoclonal Abs as expected, and also improved tolerance to Ab-mediated neutralization in vitro. These are very interesting and promising results, though the higher transduction efficiency of hAEV6 makes it difficult to attribute the neutralization results solely to higher antibody tolerance.

To demonstrate the real-life utility of the new hAEV6 capsid variant more convincingly, the authors should consider two additional experiments:

- a) perform in vitro neutralization assay with 5 different lots of IVIG and determine whether the 10-fold higher tolerance to neutralization observed with the 21 pooled monoclonal Abs can be confirmed. The prediction would be a lower IC50 neutralization titer for IVIG.
- b) Verify the predicted higher enrollment rate by using 50-100 randomly selected individual healthy or disease donor plasma samples and confirming a lower seroprevalence for hAEV6 vs. AAV9 neutralizing titers >50. Right now, the higher enrolment rate mentioned in the Discussion is more of a conjecture rather than a fact. As it stands, the observed 10-fold lower neutralizing titers could be restricted to a small number of individual Zolgensma-treated donors. I remain concerned that such small titer changes in a very limited number of subjects may not translate to larger randomly selected subject cohorts, thus precluding clinically meaningful improvements of eligibility rates. This will be important to address to convincingly show that the selected 21 monoclonal Abs are truly representative of natural human pre-existing AAV9 immunity.

Minor: Figure 7D x-axis labeling could be improved to be more comprehensive. Do you mean the molar ratio between Abs and capsid, ie. molar excess of Ab molecules over capsid particles? Please make this more clear.

The authors may also want to comment whether they expect the hAEV6 variant to be less immunogenic or whether post-administration immunity to hAEV6 would be anticipated to be similar to that of AAV9. If similar, then yet another immune escape variant would be required for repeat dose administration, resulting in an endless cycles of having to reengineer immune escape variants after each dose? Would this really be an economically feasible approach to redosing, or is the authors' approach more suitable for overcoming pre-existing AAV9 immunity? This should be considered in the Discussion.

Reviewer #3

(Remarks to the Author)

The authors structurally characterized 21 human-derived antibodies from patients treated with the AAV9 vector using high-resolution cryoEM technology. Localized reconstructions were applied to determine the structures of the bound antibodies. Overall, 21 structures with resolutions ranging from 1.88 to 3.27 Å were determined. These complex structures reveal the interactions of the mAbs binding to AAV vectors. More importantly, key residues of the capsid were identified as antibody escape mutations. Understanding the escape mechanism facilitates the future design of capsids, which can be used to treat patients who have already developed AAV9 vector resistance.

Minor:

1. In line 133, please provide a definition for "sub-particle maps."
2. In lines 159, 161, and 178, the authors described how the CDR loops interact with the AAV. It would enhance clarity to depict these interactions in a figure, illustrating all critical interactions.
3. In line 169, the assumption "possibly because it does not enter the 2-fold depression" should be discussed in the discussion section rather than in the results section.
4. In Fig. 3A, including side views to illustrate fab binding, although details are described in the manuscript, would enhance comprehension. With a figure presenting side views, readers can better understand the subclasses.
5. In Figure 2, please include a color bar indicating the radial distance from centers. Additionally, define the little pentagons and ovals in the legends.
6. In Figure 3, reorganize the legends of B, C, and D to prevent redundancy. A color bar is also necessary here.
7. In Figure 4, the legends A and B do not correspond to the figure display. The fab2-1 is depicted in purple and the AAV9 in yellow. However, in B, C, and D, all residues are colored yellow. Please ensure consistency in color usage to provide clear display for readers. Additionally, correct the language in the legend.
8. Please define "2f" listed on the y-axis in Figure 5.
9. In line 199, the authors state that "Fab1-3 had the lowest number of contacts with the AAV9 capsid, followed by Fab1-1.

The Fabs with the highest number of contacts all belong to group 2 of the 2-fold binders." However, the number of contacts alone does not determine the significance of the interaction. Bond types or buried surface area (BSA) would better indicate the significance of the contacts.

10. In Figure 6, please describe how distance is calculated, whether from Calpha or Cbeta.

11. Please define "B1" in Figure S2.

12. In line 220, please define "DE-loops," either in the figure or in the main text.

13. In Figure S5, the arrows indicating shifting are confusing.

14. In Figure S6, there is no need to color all non-carbon atoms if nothing related needs clarification. Using two different colors to show AAV9 and fabs would provide a clearer view.

15. In the method "cryoEM sample preparation and data collection," please include the blotting force and time range used.

Version 1:

Reviewer comments:

Reviewer #1

(Remarks to the Author)

The authors provided a revised version of the manuscript and a rebuttal letter. The authors have addressed the reviewer's concerns.

Since the authors identified HSPG binding as a likely reason for the enhanced infectivity of hAEV6 they refused to perform the requested biodistribution. However, it would have been interesting to see the changes in biodistribution and transduction compared to AAV9. Because of the expected changes in biodistribution, discussion has been edited and authors opted for hAEV5 as bases for further developments.

The authors state that > all approved AAV biologics utilize the capsids of naturally occurring AAVs.< The authors might need to edit this statement in light of AAVRh74var (BEQVEZ). The authors might want to check this.

Reviewer #2

(Remarks to the Author)

Previous Reviewer Comment: To demonstrate the real-life utility of the new hAEV6 capsid variant more convincingly, the authors should consider two additional experiments:

a) perform in vitro neutralization assay with 5 different lots of IVIG and determine whether the 10-fold higher tolerance to neutralization observed with the 21 pooled monoclonal Abs can be confirmed. The prediction would be a lower IC50 neutralization titer for IVIG. b) Verify the predicted higher enrollment rate by using 50-100 randomly selected individual healthy or disease donor plasma samples and confirming a lower seroprevalence for hAEV6 vs. AAV9 neutralizing titers >50. Right now, the higher enrolment rate mentioned in the Discussion is more of a conjecture rather than a fact. As it stands, the observed 10-fold lower neutralizing titers could be restricted to a small number of individual Zolgensma-treated donors. I remain concerned that such small titer changes in a very limited number of subjects may not translate to larger randomly selected subject cohorts, thus precluding clinically meaningful improvements of eligibility rates. This will be important to address to convincingly show that the selected 21 monoclonal Abs are truly representative of natural human pre-existing AAV9 immunity.

--

Response: We thank the reviewer for their suggestion. We currently do not have the quantity of patient materials available to satisfy the reviewer's comment. However, our preliminary data shows a general reduction of reactivity with the hAEV6-Q588R capsid compared to the AAV9 capsid, with few individuals having increased reactivity. This will be studied in more detail in a future study. Nonetheless, the overall trend is the same as presented by our data in the manuscript. Thus, we feel confident to extrapolate our data onto a recent serology study to predict the percentage of children that additionally may now qualify for treatment. This is now discussed in the manuscript.

--

New Reviewer Comment: While it is slightly disappointing to see my previous suggestions largely being dismissed by the authors, based on their claim about unavailable patient samples, I was profoundly excited to review the new hAEV6-neutralizaion data for 8 healthy human donors. These new data must be included in the manuscript in Fig. 7 at the very minimum, rather than solely providing them to the Reviewers in the Response Letter. - What it is not clear to me is: what exactly prevented the authors from testing at least 50 randomly selected healthy donor samples for hAEV6 neutralizing titers, as is customary in seroprevalence studies to arrive at a more stastically sound conclusion? These healthy donor serum samples can quite easily be purchased from any reputable commerical biospecimen vendor. I would much rather rely on a more powerful seroprevalance data set to support claims of a higher eligibility rate than the authors' gut feelings, ambiguous trends, and their unfounded confidence to speculate from what I would call an extremely limited sample size.

Reviewer #3

(Remarks to the Author)

I have no concerns and recommend the manuscript for publication in its revised form.

Version 2:

Reviewer comments:

Reviewer #2

(Remarks to the Author)

The authors have addressed my concerns.

Dear Editor and Reviewers,

We would like to take this time to thank you and the reviewers for their constructive comments and suggestions regarding our submitted manuscript.

Below, point-by-point we address the questions, comments and suggestions they raised. Please refer to **our yellow highlighted responses** and attached updated manuscript and figures.

Reviewer #1:

The manuscript Structural characterization of antibody-responses from Zolgensma treatment provides a blueprint for the engineering of an AAV capsid suitable for redosing submitted by Mietzsch and colleagues provides a unique insight on how human antibodies bind to AAV9 capsids.

Response: We thank the reviewer for their suggestions to improve our manuscript.

The monoclonal antibodies assayed within this study are derived from patients receiving Zolgensma. Thus, extremely high vector doses (10^{14} per kilogram body weight) had been applied to presumably very young patients which both might have an impact on the antibody pool present in the patients following gene therapy. The possible impact of those parameters should be discussed by the authors. Moreover, is the source - like in the previous publication - three infants?

Response: The reviewer is correct. The antibody structures determined in this publication are from three infants. This has been clarified in the abstract. Furthermore, we expanded our discussion to address the differences of anti-AAV9 antibody titers in patients following SMA gene therapy and individuals with naturally acquired anti-AAV9 antibodies.

According to the knowledge of the reviewer, six and not seven AAV gene therapies have received market approval. The authors might edit the introduction accordingly.

Response: The number of seven approved AAV gene therapies was indeed correct (Glybera, Luxturna, Zolgensma, Roctavian, Hemgenix, Upstaza, Elevidys, Beqvez) but since then has increased to eight. This has been updated in the introduction.

The AAV capsid is composed of 60 subunits to which three VPs contribute. This is different from the authors' statement on 60 copies of the viral proteins. The authors need to revise their statement.

Response: The statement has been revised and reads now: "The AAV vectors are composed of non-enveloped T=1 icosahedral capsids, consisting of 60 viral proteins (VP)"

One caveat of this very interesting study is that it remains unclear which of the antibodies are neutralizing antibodies. This is of high interest as authors report on differences in the structures recognized by murine compared to human antibodies AND on conformational changes of the capsid upon antibody binding. This information must be included.

Response: All the human antibodies neutralize AAV9 transduction. This information was provided in our previous publication and for clarity this information has been now added to the manuscript.

The authors cloned a new capsid with six aa changes (T491R, D556P, N562Y, T582Q, Y706D). Why were these changes and not additional or other residues identified in their study used? On which bases were the aa substitution chosen. When describing Fab binding aa T584 is mentioned. The new variant contains a T582Q substitution. Is there a mistake in the numbering?

Response: We apologize, T584 was indeed a typographical error and has been corrected to T582. The aa substitutions were chosen as stated: "Amino acid substitutions of the identified contact residues were selected either by disrupting critical interactions of the Fab with the capsid or by introducing steric clashes, preventing antibody binding (Fig.S6). The newly generated capsids were analyzed for their productivity and infectivity to exclude non-viable variants. Subsequently, these variants were tested for their ability to escape the mAbs." Additional information has been added to the manuscript stating: "Successful aa substitutions were combined to generate an AAV9 capsid variant with the fewest changes capable of evading multiple antibodies."

The authors did not include aa substitutions for residues at the 5-fold and 2/5-fold wall because they are non-infections. However, Table S5 is lacking details on how infectivity was defined.

Response: A footnote stating "Non-infectious capsid variants showed transduction efficiencies <10% vs. wtAAV9 capsids" has been added to Table S5.

In the presence of antibodies, hAEV6 shows an increased transduction efficiency. How is this explained?

Response: A slight enhancement of infection up to ~2-fold of AAV vectors in presence of antibodies that are unable to neutralize transduction has been observed previously *in vitro*. The reason for this is currently unknown. As these antibodies are unable to bind to the capsids, they might affect the cells. The antibodies might bind to cell surface Fc receptors and indirectly affect intracellular trafficking of the AAV vectors. Additionally, as antibodies are glycosylated, this may modulate the transduction process in some way. References to previous studies mentioning these observations have been added to the manuscript.

hAEV6 and AAV9 differ in infectivity. How do the authors explain the higher infectivity of hAEV6 and was this difference considered when performing the neutralization assay? This is crucial to state that the hAEV6 is indeed an immune escape variant.

On the same line, authors state in the final paragraph that >While none of the changes in the hAEV6 variant targeted these antibodies, surprisingly, a shift in the neutralization assay was observed, allowing this capsid variant to tolerate ~10-fold higher antibody concentrations before being neutralized. This effect could be associated with the higher transduction efficiency of the hAEV6 variant. If this is indeed the case, the question on the value of the antibody binding study for the reported capsid engineering arises.

Response: We have conducted additional experiments. The increased transduction efficiency is the result of the substitution of glutamine to arginine in position 588 (Q588R). The capsid without this amino acid change (hAEV5) is lacking the enhancement of infection (Figure 7C). The arginine

resembles the arginines of AAV2 in VR-VIII and conferred the ability of the hAEV6-Q588R variant to bind heparin/HSPG as shown by heparin competition assay (Figure 7D).

Moreover, is there a difference in empty-to-full capsid ratio between hAEV6 and AAV9? This is important since antibodies bind the capsid independent on whether they contain a vector genome or not.

Response: To address the reviewer's comment, we loaded equivalent amounts of vectors (hAEV5 and AAV9) based on their genome titer on an SDS-PAGE. This gel shows similar VP band intensities which indicates that the empty-to-full capsid ratio is comparable (Figure S7)

Since multiple changes to the capsid were made to generate hAEV6, a biodistribution study needs to be performed. Infectivity data are so far limited to a single cell line, which is highly permissive for several serotypes. Since authors state that hARV6 with its improved infectivity might be used as an alternative to AAV9, the mentioned biodistribution study is required.

Response: We have conducted additional experiments to address the reviewer's comment. A biodistribution of the hAEV5 capsid variant was conducted in C57BL/6 mice. The data is presented in Figure 7G and Figure S9. Our data shows that the 5 amino acid substitutions of hAEV5 do not affect the overall biodistribution of the capsid variant. The biodistribution was not analyzed for the HSPG-binding hAEV6 capsid. Due to the ubiquitous expression of HSPG in most tissues the biodistribution will likely be changed compared to AAV9. A discussion on this issue has been added to the manuscript.

Reviewer #2:

Mietzsch et al. characterize the structure of several human-derived monoclonal Abs with AAV9 capsids, suggesting that many of them bind by inducing conformational changes. Mutagenesis performed with the AAV9 capsid to disrupt Ab epitopes indicates the path towards new capsid variants that may escape pre-existing or treatment-emergent humoral immune responses. I am not able to comment on the validity of processing structural data using localized reconstructions and symmetry relaxation and will defer to other reviewers who are experts in this field. The empirical basis for induced conformational changes should also be carefully evaluated by these structural experts, since I am unclear what the quantitative threshold for such change might be, as compared to simple "noise" in the cryo-EM data generated for the Ab-capsid complexes, or whether the observed conformational changes were an artefact of the localized reconstruction models.

Response: The localized reconstruction protocol was used to determine the 2-, 3-, or 5-fold region by itself, independent of the entire capsid. The obtained resolution and experimental derived high quality of the maps exclude the possibility that the conformational changes are noise. Furthermore, the majority of the AAV9 capsid structure of the sub-particle map matches with previous reported AAV9 capsid structures. This information was added to the manuscript.

Nonetheless, the derived models for the complexes predicted that individual amino acids on the capsid surface are involved in Ab binding, which was tested by targeted mutagenesis. The newly generated AAV9 capsid variant (hAEV6) carries 6 aa changes and eliminates binding to 18 of the 21 monoclonal Abs as expected, and also improved tolerance to Ab-mediated neutralization in vitro. These are very interesting and promising results, though the higher transduction efficiency of hAEV6 makes it difficult to attribute the neutralization results solely to higher antibody tolerance.

Response: We thank the reviewer for their positive comments on our manuscript.

To demonstrate the real-life utility of the new hAEV6 capsid variant more convincingly, the authors should consider two additional experiments:

- a) perform in vitro neutralization assay with 5 different lots of IVIG and determine whether the 10-fold higher tolerance to neutralization observed with the 21 pooled monoclonal Abs can be confirmed. The prediction would be a lower IC50 neutralization titer for IVIG.
- b) Verify the predicted higher enrollment rate by using 50-100 randomly selected individual healthy or disease donor plasma samples and confirming a lower seroprevalence for hAEV6 vs. AAV9 neutralizing titers >50. Right now, the higher enrolment rate mentioned in the Discussion is more of a conjecture rather than a fact. As it stands, the observed 10-fold lower neutralizing titers could be restricted to a small number of individual Zolgensma-treated donors. I remain concerned that such small titer changes in a very limited number of subjects may not translate to larger randomly selected subject cohorts, thus precluding clinically meaningful improvements of eligibility rates. This will be important to address to convincingly show that the selected 21 monoclonal Abs are truly representative of natural human pre-existing AAV9 immunity.

Response: We thank the reviewer for their suggestion. We currently do not have the quantity of patient materials available to satisfy the reviewer's comment. However, our preliminary data shows a general reduction of reactivity with the hAEV6-Q588R capsid compared to the AAV9 capsid, with few individuals having increased reactivity. This will be studied in more detail in a future study. Nonetheless, the overall trend is the same as presented by our data in the manuscript. Thus, we feel confident to extrapolate our data onto a recent serology study to predict the percentage of children that additionally may now qualify for treatment. This is now discussed in the manuscript.

Minor: Figure 7D x-axis labeling could be improved to be more comprehensive. Do you mean the molar ratio between Abs and capsid, ie. molar excess of Ab molecules over capsid particles? Please make this more clear.

Response: The x-axis was changed to molar ratio of mAbs to capsids and is now also described in the figure legend.

The authors may also want to comment whether they expect the hAEV6 variant to be less immunogenic or whether post-administration immunity to hAEV6 would be anticipated to be similar to that of AAV9. If similar, then yet another immune escape variant would be required for repeat dose administration, resulting in an endless cycles of having to reengineer immune escape variants after each dose? Would this really be an economically feasible approach to redosing, or is the authors' approach more suitable for overcoming pre-existing AAV9 immunity? This should be considered in the Discussion.

Response: The objective of this study is not to make AAV capsids that are less immunogenic. Our variants, as any other engineered AAV capsid, will not prevent future immune responses to the modified capsid but may expand the patient cohort treatable with AAV vectors that previously have been excluded due to their pre-existing antibodies. This point has been made clearer in the discussion.

Reviewer #3:

The authors structurally characterized 21 human-derived antibodies from patients treated with the AAV9 vector using high-resolution cryoEM technology. Localized reconstructions were applied to determine the structures of the bound antibodies. Overall, 21 structures with resolutions ranging from 1.88 to 3.27 Å were determined. These complex structures reveal the interactions of the mAbs binding to AAV vectors. More importantly, key residues of the capsid were identified as antibody escape mutations. Understanding the escape mechanism facilitates the future design of capsids, which can be used to treat patients who have already developed AAV9 vector resistance.

Response: We thank the reviewer for their positive comments on our manuscript.

Minor:

1. In line 133, please provide a definition for "sub-particle maps."

Response: A more detailed description is now provided in line 132-134. "Thus, the localized reconstruction method was used in combination with symmetry relaxation to structurally characterize the 2-, 3-, or 5-fold regions, independent of the whole capsid, to reconstruct the Fabs not conforming to the icosahedral symmetry of the capsid. In this process sub-particle maps are generated comprising only the region around the symmetry axis of choice."

2. In lines 159, 161, and 178, the authors described how the CDR loops interact with the AAV. It would enhance clarity to depict these interactions in a figure, illustrating all critical interactions.

Response: In this study the interaction of 21 Fabs to the AAV9 capsid is analyzed. Thus, not all interaction can be shown in figures. Figure 4D and S6 show some exemplary interactions between the capsid and the Fab. The remaining interactions are listed in Tab.S2-S4. Additionally, all the maps and models are deposited to the electron microscopy databank and can be downloaded and viewed.

3. In line 169, the assumption "possibly because it does not enter the 2-fold depression" should be discussed in the discussion section rather than in the results section.

Response: The assumption was removed from the results section. The 2-fold binding antibodies in relation to the CDR3 length and their ability to enter the 2-fold depression is discussed in the discussion section.

4. In Fig. 3A, including side views to illustrate fab binding, although details are described in the manuscript, would enhance comprehension. With a figure presenting side views, readers can better understand the subclasses.

Response: We initially also thought that the side-views would be better to visualize the different classes. However, it turned out that side-views (as in Figure 4A) result in too much overlapping of the chains, making it almost impossible to distinguish between the light and heavy chains (regardless of the colors used) and therefore understanding the different class groupings.

5. In Figure 2, please include a color bar indicating the radial distance from centers. Additionally, define the little pentagons and ovals in the legends.

Response: A reference to the scale (color) bar in Figure 1 is provided and all the symbols described in the legend.

6. In Figure 3, reorganize the legends of B, C, and D to prevent redundancy. A color bar is also necessary here.

Response: The figure legend was reorganized, and a color bar was added to the figure.

7. In Figure 4, the legends A and B do not correspond to the figure display. The fab2-1 is depicted in purple and the AAV9 in yellow. However, in B, C, and D, all residues are colored yellow. Please ensure consistency in color usage to provide clear display for readers. Additionally, correct the language in the legend.

Response: The figure legend was corrected and the color of the CDR models changed for consistency as the reviewer suggested.

8. Please define "2f" listed on the y-axis in Figure 5.

Response: "2f" is now defined as '2-fold' in the figure legend.

9. In line 199, the authors state that "Fab1-3 had the lowest number of contacts with the AAV9 capsid, followed by Fab1-1. The Fabs with the highest number of contacts all belong to group 2 of the 2-fold binders." However, the number of contacts alone does not determine the significance

of the interaction. Bond types or buried surface area (BSA) would better indicate the significance of the contacts.

Response: The reviewer is correct that the number of interactions alone does not determine if an antibody is good binder. However, there is a moderate correlation between the number of interactions and interaction/buried surface area ($R^2 = 0.63$). We added the information on the interaction surface area to the paragraph.

10. In Figure 6, please describe how distance is calculated, whether from Calpha or Cbeta.

Response: The distances are measured in Coot between the same atom of the same amino acid for the two models. This information was added to the figure legend.

11. Please define "B1" in Figure S2.

Response: The reviewer likely refers to Figure S3. A description of B1 has been added to the figure legend: The mAb B1 is used as a control which only detects denatured AAV capsids.

12. In line 220, please define "DE-loops," either in the figure or in the main text.

Response: DE-loops was replaced with VR-II.

13. In Figure S5, the arrows indicating shifting are confusing.

Response: The arrows in Figure S5 were removed.

14. In Figure S6, there is no need to color all non-carbon atoms if nothing related needs clarification. Using two different colors to show AAV9 and fabs would provide a clearer view.

Response: We followed the advice of the reviewer to color the carbons of the AAV9 and Fab model differently but kept the coloring of the non-carbon atoms.

15. In the method "cryoEM sample preparation and data collection," please include the blotting force and time range used.

Response: The information was added to the method section.

REVIEWER COMMENTS

Reviewer #1:

The authors provided a revised version of the manuscript and a rebuttal letter. The authors have addressed the reviewer's concerns.

Since the authors identified HSPG binding as a likely reason for the enhanced infectivity of hAEV6 they refused to perform the requested biodistribution. However, it would have been interesting to see the changes in biodistribution and transduction compared to AAV9. Because of the expected changes in biodistribution, discussion has been edited and authors opted for hAEV5 as bases for further developments.

The authors state that > all approved AAV biologics utilize the capsids of naturally occurring AAVs.< The authors might need to edit this statement in light of AAVRh74var (BEQVEZ). The authors might want to check this.

Response: We thank the reviewer for his comments that enabled us to improve our manuscript. Prior to the reviewer's comment, we were under the impression that Pfizer's approved gene therapy product (Beqvez) was using the wild-type AAVrh.74 capsid. Thus, we initiated a search for additional information on the capsid used. However, our findings were limited, as the sequence of the capsid is not publicly available. So, we reached out to prominent researchers in the AAV field, more familiar with Beqvez, and were informed that Pfizer is using a natural occurring AAVrh.74 capsid variant (to partially to circumvent Sarepta's patent on their rh74 gene therapy, Elevidys). This variant differs from AAVrh74 in four amino acids located within the structural disordered VP1/2 common region, so not affecting the previously characterized basic regions. Thus, our statement in the manuscript "To this date, all approved AAV biologics utilize the capsids of naturally occurring AAVs" is still correct. As the AAVrh74 capsid is not the topic of this study we would prefer not to add the above information to our manuscript.

Reviewer #2:

While it is slightly disappointing to see my previous suggestions largely being dismissed by the authors, based on their claim about unavailable patient samples, I was profoundly excited to review the new hAEV6-neutralization data for 8 healthy human donors. These new data must be included in the manuscript in Fig. 7 at the very minimum, rather than solely providing them to the Reviewers in the Response Letter. What is not clear to me is: what exactly prevented the authors from testing at least 50 randomly selected healthy donor samples for hAEV6 neutralizing titers, as is customary in seroprevalence studies to arrive at a more statistically sound conclusion? These healthy donor serum samples can quite easily be purchased from any reputable commercial biospecimen vendor. I would much rather rely on a more powerful seroprevalence data set to support claims of a higher eligibility rate than the authors' gut feelings, ambiguous trends, and their unfounded confidence to speculate from what I would call an extremely limited sample size.

Response: We did not think this was necessary for our manuscript, but are happy to include the data provided in the rebuttal letter into the manuscript. This data was generated by a collaborator who is now a co-author on this manuscript (Nicholas JC Smith). Following the

reviewer's comments, we have discussed this data in detail with them. The data generated is based on 50 sera of healthy individuals. Several studies have shown that the prevalence of anti-AAV9 antibodies in the general population can range between 5-50%. For this set of random healthy donors from Australia 8 out of 50 (16%) individuals possessed antibodies against the AAV9 capsid, whereas the remaining 42 sera were below the detection limit. While most of these sera do not react with either the AAV9 or our engineered capsid, one showed minor reactivity to the hAEV6 capsid (at the lowest dilution). Thus, only the data on the 9 seropositive individuals is relevant and has been included in this study (Figure S9A). Additionally, the data includes the analysis of sera from four individuals that were previously precluded from receiving treatment with an AAV9-mediated gene vector. The description of the data has been added to the result section (line 295-302).

Reviewer #3:

I have no concerns and recommend the manuscript for publication in its revised form.

Response: We thank the reviewer for his comments that enabled us to improve our manuscript.